# Elevated cytokines and chemokines in peripheral blood of patients with SARS-CoV-2 pneumonia treated with high-titer convalescent plasma

**Stacey L. Fanning**[1‡]*, **Robert Korngold**[2,3‡], **Zheng Yang**[2], **Kira Goldgirsh**[2], **Steven Park**[2], **Joshua Zenreich**[3], **Melissa Baker**[3], **Phyllis McKiernan**[3], **Ming Tan**[4], **Bingsong Zhang**[4], **Michele L. Donato**[3], **David S. Perlin**[2]

**1** Touro College of Osteopathic Medicine, New York, New York, United States of America, **2** Hackensack Meridian Health Center for Discovery and Innovation, Nutley, New Jersey, United States of America, **3** John Theurer Cancer Center, Hackensack University Medical Center, Hackensack New Jersey, United States of America, **4** Department of Biostatistics, Bioinformatics & Biomathematics, Georgetown University, Washington DC, United States of America

‡ Co-first authors.
* stacey.fanning@touro.edu

**Data Availability Statement:** All relevant data are within the manuscript and its Supporting Information files.

## Abstract

The global SARS-CoV-2 coronavirus pandemic continues to be devastating in many areas. Treatment options have been limited and convalescent donor plasma has been used by many centers to transfer passive neutralizing antibodies to patients with respiratory involvement. The results often vary by institution and are complicated by the nature and quality of the donor plasma itself, the timing of administration and the clinical aspects of the recipients. SARS-CoV-2 infection is known to be associated with an increase in the blood concentrations of several inflammatory cytokines/chemokines, as part of the overall immune response to the virus and consequential to mediated lung pathology. Some of these correlates contribute to the cytokine storm syndrome and acute respiratory distress syndrome, often resulting in fatality. A Phase IIa clinical trial at our institution using high neutralizing titer convalescent plasma transfer gave us the unique opportunity to study the elevations of correlates in the first 10 days after infusion. Plasma recipients were divided into hospitalized COVID-19 pneumonia patients who did not (Track 2) or did (Track 3) require mechanical ventilation. Several cytokines were elevated in the patients of each Track and some continued to rise through Day 10, while others initially increased and then subsided. Furthermore, elevations in MIP-1α, MIP-1β and CRP correlated with disease progression of Track 2 recipients. Overall, our observations serve as a foundation for further study of these correlates and the identification of potential biomarkers to improve upon convalescent plasma therapy and to drive more successful patient outcomes.

**Funding:** This work was supported by the COVID Emergency Research Fund #61315, Hackensack University Medical Center to MLD and by funds provided to the Hackensack Meridian Health Center for Discovery and Innovation via DSP by Activision Publishing Inc, Suez North America (https://www.suez-na.com/en-us/who-we-are/suez-in-north-america/charitable-giving), and NJ Stands Up to COVID (http://give.hackensackmeridianhealth.org/site/TR/DIY/DIYFundraising?px=1418344&pg=personal&fr_id=1860). The funders of this study had no role in the study design, data collection, data analysis, data interpretation, or writing of the report.

**Competing interests:** The authors have declared that no competing interests exist.

## Author summary

COVID-19, the disease caused by the SARS-CoV-2 virus, has a varied clinical course with limited treatment options. While some patients mount a productive immune response leading to recovery, others progress to rapid respiratory deterioration that may require hospitalization and mechanical ventilation. Our institution conducted a clinical trial to evaluate the efficacy of convalescent plasma therapy (CPT) to treat patients hospitalized with COVID-19 pneumonia. In this arm of the study, we sought to examine immune analytes in donor plasma as well as evaluate the recipients' plasma before CPT infusion, and at Day 3 and Day 10 post-CPT infusion. We found some analytes to be elevated in plasma donors, compared to healthy controls, even after recovery. Plasma composition in CPT recipients prior to infusion showed elevations in several analytes associated with immune activation. Some significant differences were seen in plasma composition in patients in our Track 2 cohort (hospitalized without mechanical ventilation) compared to the Track 3 cohort (hospitalized with mechanical ventilation). In addition, we obtained plasma samples for hospitalized COVID-19 patients that did not receive CPT and noted several differences in the course of immune analyte production over time compared to the CPT-treated patients.

## Introduction

Since the emergence of SARS-CoV-2 as a human pathogen in December of 2019 and its global spread resulting in a worldwide pandemic, scientists and clinicians have been working non-stop to understand the biology of the virus and the ensuing host response to discover treatments for COVID-19 disease. The clinical course of COVID-19 is complex and diverse ranging from asymptomatic or mild infections to more severe cases requiring hospitalization and mechanical ventilation with some progressing to respiratory failure and death. Damage to organ systems in severe infection is thought to be mediated by both the virus and the concomitant host immune response [1,2]. The SARS-CoV-2 virus enters angiotensin-converting enzyme 2 (ACE2) receptor-expressing cells using its spike protein [3], and viral entry into host cells triggers a robust anti-viral inflammatory response [2]. The inflammatory response to SARS-CoV-2 and resultant cytokine storm is believed to account for much of the severe disease pathology in later stages of disease leading to acute respiratory distress syndrome (ARDS) and multi-organ failure [2,4,5]. Early reports have described elevations in a variety of cytokines and inflammatory markers with correlations to disease severity [2,4,6]. However, cytokine storm syndrome (CSS) remains ill-defined. Despite the early association of the cytokine response to SARS-CoV-2 infection with CSS, recent reports have indicated that COVID-19 cytokine profiles are not consistent with CSS as characterized in patients with sepsis [7]. Other reports have shown that the cytokine profile of COVID-19 patients is distinct from that seen in other respiratory viral infections such as influenza [8]. Understanding the course of cytokine production and inflammatory markers throughout progression of disease and their contribution to respiratory failure will be critical to optimizing treatment protocols. In addition, determining early predictors of worsening disease will aid in treatment decisions, particularly in the use of immunosuppressants at critical junctures during the clinical course.

Convalescent plasma transfer (CPT) serves as a viable treatment option for emerging infectious diseases with limited approved pharmacologic therapies and/or vaccines. CPT allows for the immediate effect of pathogen-specific passive immunity from donor to recipient via circulating Ig. Several studies have demonstrated CPT to be safe for use and effective at transferring

anti-viral immunity to patients hospitalized with SARS-CoV-2 pneumonia [9–16]. A national program was established and implemented nationwide, but it has been met with mixed success [17,18]. When used successfully, the timing and high threshold level of neutralizing antibodies in recovered sera used for therapy are critical factors [19,20]. Our institution conducted a phase IIa prospective study which aimed to determine the intubation rate, survival, viral clearance, and the development of endogenous antibodies in patients with COVID-19 pneumonia treated with convalescent plasma containing high levels of neutralizing anti-SARS-CoV-2 antibodies [21]. Of the 51 CPT treated hospitalized patients, those initially not on mechanical ventilation (Track 2) exhibited a significantly improved day-30 survival of 88.9% compared to non-CPT treated COVID-19 patients* of 72.5% (p = 0.036) (* non-CPT COVID-19 patients treated within our hospital network and retrospectively identified using electronic health records). Those patients initially hospitalized with mechanical ventilation (Track 3) had a day-30 mortality of 46.7%, compared to an equivalent non-CPT treated group with a mortality of 71% (p = 0.08)[21].

Ancillary to our clinical trial, we evaluated the cytokine and Ig isotype composition of convalescent plasma derived from donors as well as the plasma from CPT recipients at times prior to infusion, and at Day 3 and Day 10 post-infusion. Here, we provide the first comprehensive immunologic assessment of recipient patients following CPT. The patterns of thirty-five elevated cytokines, chemokines and inflammatory markers are profiled in these patients, including dominant ones that were elevated in over 70% of the recipients, such as IL-6, IL-7, IL-8, interferon gamma-induced protein 10 (IP-10/CXCL10), monocyte chemoattractant protein-1 (MCP-1/CCL2), macrophage inflammatory protein-1beta (MIP-1β /CCL4), vascular endothelial growth factor (VEGF), c-reactive protein (CRP), and neutrophil gelatinous-associated lipocalin (NGAL). In addition, immunoglobulin isotype analysis was performed and the observation of Ig isotype class switching over time towards IgG4 and IgA in both Track 2 and Track 3 patients was noted.

Identifying biomarkers that can better predict successful therapeutic responses will improve outcomes, especially as immune-escape viral variants are becoming more prevalent.

## Materials & methods

### Ethics statement

We conducted a single institution prospective phase IIa clinical trial, registered with Clinical-Trials.gov NTC04343755, with FDA IND approval obtained 4/4/2020 and approved by our Institutional Review Board. Written informed consent was obtained from all donors and recipients.

### Study design

For research purposes across studies, patients with COVID-19 at our institution were divided into three tracks based on acuity, Track 1 being attributed to outpatients, Track 2 for patients hospitalized but not requiring positive pressure mechanical ventilation, and Track 3 for patients receiving positive pressure mechanical ventilation. The present study was limited to patients in Track 2 and Track 3. The goal of the trial was to determine the intubation rate, survival, viral clearance, and the development of endogenous antibodies in patients with COVID-19 pneumonia treated with convalescent plasma containing high levels of neutralizing anti-SARS-CoV-2 antibodies The clinical results of this trial which involved 51 patients that received donor plasma have been published [21], and the correlative science study, reported herein, investigated the levels of peripheral blood cytokines, chemokines, and Ig isotypes from the first 35 patients enrolled in the original study.

## Convalescent plasma donors

Prospective plasma donors were included if they were aged 18 to 60 years, had a history of a positive nasopharyngeal swab for SARS-CoV-2 or a positive antibody test, were at least 14 days from resolution of symptoms, had one subsequent negative swab, were found to have high-titers of neutralizing antibodies against SARS-CoV-2 (>1:500), and met institutional and FDA regulations for donation of blood products. Participants completed a health questionnaire, were given a physical examination, had blood analysis performed for complete cell count and chemistry, infectious disease markers, and the presence of anti-HLA antibodies for female donors. The presence of SARS-CoV-2 neutralizing antibodies in volunteer donors was evaluated using the previously described SARS-CoV-2 ELISA protocol with recombinant spike receptor binding domain (RBD) as capture antigen [22]. Antibody titer from recovered donors was evaluated as described elsewhere [21].

## Patient population

Patient inclusion criteria required age 18 years or older and hospitalization for the management of symptoms associated with a documented infection with SARS-CoV-2. Patients were excluded for a history of severe transfusion reactions, infusion of Ig within 30 days, AST or ALT greater than 10 times the upper limit of normal, or requirement for vasopressors and dialysis. Patients requiring intermittent vasopressors for sedation management were treated.

Cryopreserved plasma samples from hospitalized COVID-19 patients who did not receive CPT were obtained from the institutional biorepository. Samples were selected based upon time from symptom onset that correlated with sample collection times in our CPT cohorts. For the Track 2 CPT cohort, the average time from symptom onset until the pre-infusion collection was 10 days. Therefore, in the nonCPT Track 2 cohort we examined samples collected at or around 10 days after symptom onset as well as those collected approximately 10 days later to correspond to the Day 10 post-infusion collection time point in the CPT cohort. For the Track 3 CPT cohort, the average time from symptom onset to infusion was 15 days. Therefore, in the nonCPT Track 3 cohort, we examined samples collected at or around day 15 and again 10–15 days later. As these samples were obtained retroactively, detailed patient characteristics were not readily available.

## Plasma collection and infusion procedure

Donors underwent plasmapheresis using the Trima Accel system for either a planned fresh infusion of 500 mL or for cryopreservation in aliquots of 200 mL. Recipients were administered a single infusion of convalescent plasma (fresh or frozen) at a rate less than 250 mL per hour. Premedication with diphenhydramine 25 mg IV and hydrocortisone 100 mg IV with or without acetaminophen was given. The use of fresh versus frozen plasma, based solely on the availability of product at the time of request, had no impact on clinical outcome [21]. Exploratory blood work including serology for anti-SARS-CoV-2 titers was performed immediately pre-infusion and on Day 3, and Day 10 post-treatment. Samples were taken from all living patients on Day 10 regardless of hospitalization status. SARS-CoV-2 testing by RT-PCR from nasopharyngeal or endotracheal tube secretions was done on Day 10. A 10 mL sample of plasma was collected at the bedside from the donor plasma bag immediately pre-infusion for analysis. Further details of collection and infusion are as previously described [21].

## Cytokine and biomarker quantification

Donor plasma and recipient plasma pre-infusion, Day 3 post-infusion and Day 10 post-infusion samples collected for cytokine/biomarker analysis were frozen at -80˚C upon receipt and

thawed immediately prior to assay. Thirty cytokines (IL-1RA, IL-1β, IL-3, IL-4, IL-5, IL-6, IL-7, IL-8, IL-10, IL-12p40, IL-12p70, IL-13, IL-15, IL-17A, VEGF-A, IP-10, MCP-1, MIP-1α, MIP-1β, RANTES, TNF-α, TNF-β, EGF, Eotaxin, G-CSF, GM-CSF, IFN-α2, IFN-γ, IL-1α, IL-2), antibody isotypes (IgM, IgG1, IgG2, IgG3, IgG4, IgA), C-reactive protein (CRP), NGAL (lipocalin-2), and mannose binding lectin (MBL) were measured using the Milliplex MAP assays on the Luminex xMAP platform. Assays were performed according to manufacturer's instructions. Briefly, plasma samples (neat or diluted as specified) were mixed with antibody-labeled magnetic beads in a 96-well plate and incubated with gentle shaking at room temperature for 1–2 hours as specified by the manufacturer's protocol. Following incubation, the plate was washed using a hand-held magnet for 96-well plate. Plates were incubated with detection antibody for 1 hour at room temperature with gentle shaking. Streptavidin-PE was then added and incubation proceeded for 30 minutes further. The plates were read on a Luminex-200 and analyzed with Belysa Analysis Software (Millipore Sigma). All samples were run in duplicate.

Procalcitonin (PCT) and surfactant protein-D (SP-D) were measured by ELISA using kits purchased from Millipore Sigma. Briefly, plasma samples were added to a 96-well antibody-coated ELISA plate and incubated for 2.5 hours at room temperature with gentle shaking. After washing, biotinylated detection antibody was added to each well and incubated as above for 1 hour followed by the addition of HRP-streptavidin for 45 minutes. The plates were developed by the addition of TMB reagent. Microplate reader was used to measure absorbance at 450nm. All samples were run in duplicate.

Plasma samples from 12–16 healthy adult donors purchased from the New York Blood Center (New York City, NY) prior to December 2019 to avoid the possibility of SARS-CoV-2 infection were used as controls. Elevations of correlates were defined as values greater than 2 standard deviations from the mean of the healthy controls. Some control samples with a z-score >3 were deemed outliers and eliminated. For CRP and MBL, outliers were determined by established clinical reference ranges. For CRP, control samples with values >10 mg/L were eliminated as outliers, and for MBL, values <50 ng/ml were removed.

## Statistical analysis

The primary endpoint for Track 2 was progression to positive pressure mechanical ventilation, and for Track 3 was the incidence of 30-day mortality. The decision to put Track 2 patients on mechanical ventilation was a clinical decision from the intensive care team based on multiple standard parameters that pointed to poor respiratory function (eg., oxygen levels, respiratory muscle fatigue, excessive work of breathing, etc). This decision was made independently from the research study team in order to minimize bias. The objective of this study was to investigate the association between the change of biomarker analyte (cytokines, chemokines, inflammatory markers and Ig antibody isotypes) levels between the pre-infusion time point and Day 3 or Day 10 post-infusion and corresponding endpoints of Track 2 and 3. Descriptive statistics were used to characterize the baseline profile of the subjects and biomarkers (level and elevation). The calculation of Mean (SD) were used for continuous variables; frequency and percentages were used for categorical variables. The T test was used for pairwise comparison of biomarker levels at different time points and across Tracks. To test the change of proportion of elevation between time points, the Fisher exact test was used. The overall trend of proportion change was tested by the Cochran-Armitage trend test. Logistic regression was used to identify the risk biomarkers that associated with the endpoint in each Track. P-values less than 0.05 were considered significant. Unadjusted P-values were reported for multiple comparison. Statistical analysis was completed using R software.

## Results

### Donor demographics and clinical characteristics

The demographics and characteristics of the plasma donors utilized for the first 35 recipients in this inflammatory cytokine and chemokine study are summarized in **Table 1**. The median age of the donors was 49, 27 were male and 8 were female. All donors had previously tested positive for SARS-CoV-2 infection, and among this group, 18 had relatively mild disease courses and were not hospitalized, while 5 were hospitalized and, of those, one donor was intubated. It was not known if the remaining 12 donors were hospitalized. The time of collection for plasma from the end of infection symptoms ranged from 16–44 days, with a median of 28 days. All selected plasma donors were found to have neutralizing IgG antibodies directed against the SARS-CoV-2 spike receptor binding domain (RBD) with titers ranging between 1:500 to ≥1:10,000. The majority of donor plasma were collected by apheresis and infused fresh and nine plasmas were collected and cryopreserved before use. All donor plasma samples (10 ml) for biomarker analyses were obtained directly from the infusion bags and cryopreserved before they were assayed. There was no significant difference in the percentage of patients in either track receiving different infusion titers; 26% of patients in Track 2 and 25% of patients in Track 3 received titers >10,000, 69.5% of patients in Track 2 and 66.6% of patients in Track 3 received titers of 1,000–10,000, 4.3% of Track 2 patients and 8.3% of Track 3 patients received titers of 500–1,000. No difference in clinical endpoints was detected within the ranges of donor IgG antiviral titers used as previously reported [21].

### Cytokine/chemokine concentrations and antibody composition of donor plasma

Donor plasma samples underwent multiplex assay analysis on the Luminex magnetic-bead platform, focusing on 33 biomarker analytes typically involved in inflammatory responses, with additional ELISA analyses performed for procalcitonin (PCT) and surfactant protein-D (SP-D). Analytes were considered elevated if they had concentrations above the mean + 2 standard deviations of the set of control plasma samples from healthy adult donors. The individual concentrations of analytes which were found elevated (highlighted in gray) in at least 20% of the donors are listed in **Table 2**. This group included IFN-α2 (20.0%), IL-6 (31.4%), PCT (31.4%), and CRP (25.7%), and are shown graphically in **Fig 1A** along with nine other analytes that exhibited any level of elevation. The individual concentrations of all analytes with less than 20% elevation within the donor group are detailed in the **S1 Table**.

Multiplex Luminex analysis of the concentration of specific Ig isotypes in the donor plasma samples is indicated in **Table 2** and summarized in **Fig 1B**. Of note, we found that 34.3% of the donors had elevated levels of IgG subclass IgG4. There was no significant correlation between either the elevated cytokines or Ig isotypes in the donor plasma with either progression of recipients in Track 2 towards intubation or in Track 3 towards 30-day mortality.

### Patient demographics and clinical characteristics

Patients (n = 35) were enrolled between April 15 and June 16, 2020, all with a documented infection with SARS-CoV-2 virus and with radiographic evidence of pneumonia. Demographic and baseline clinical characteristics of convalescent plasma recipients are summarized in **Table 3**. Although similar in nature to the original study [21] which involved a larger group of patients, this subset of recipients had their own distinct attributes. Patients ranged from 27 to 85 years of age (54.0, median), with 18 females and 17 males, and their BMI ranged between 20–48 (29.0, median). Race varied as indicated. Among the 35 patients analyzed in this arm of

**Table 1. Donor Characteristics.**

| Donor | Age | Gender | Hospitalized | Intubated | Time of Collection from End of Symptoms | RBD IgG detection: fold serum dilution | Plasma at Infusion— Fresh or Frozen | Infusion volume (mL) |
|---|---|---|---|---|---|---|---|---|
| Don01 | 53 | M | No | No | 27 | 1,000–10,000 | Fresh | 500 |
| Don02 | 53 | M | Unknown | Unknown | 16 | **500–1,000** | Fresh | 500 |
| Don03 | 59 | M | No | No | 25 | >10,000 | Fresh | 500 |
| Don04 | 38 | M | Unknown | Unknown | 23 | 1,000–10,000 | Fresh | 500 |
| Don05 | 60 | M | Unknown | Unknown | 21 | 1,000–10,000 | Fresh | 500 |
| Don06 | 33 | M | No | No | 28 | >10,000 | Fresh | 500 |
| Don07 | 30 | F | No | No | 29 | >10,000 | Frozen | 400 |
| Don08 | 52 | M | No | No | 28 | >10,000 | Frozen | 400 |
| Don09 | 23 | M | Unknown | Unknown | 26 | 500–1,000 | Fresh | 500 |
| Don10 * | 59 | F | No | No | 27 | 1,000–10,000 | Frozen | 400 |
| Don11 * | 59 | F | No | No | 27 | 1,000–10,000 | Fresh | 500 |
| Don12 | 49 | M | Unknown | Unknown | 23 | >10,000 | Fresh | 500 |
| Don13 | 55 | M | No | No | 29 | 1,000–10,000 | Fresh | 500 |
| Don14 | 43 | M | Unknown | Unknown | 22 | 1,000–10,000 | Fresh | 500 |
| Don15 | 59 | M | Unknown | Unknown | 26 | 1,000–10,000 | Fresh | 500 |
| Don16 | 47 | M | Yes | No | 24 | >10,000 | Fresh | 500 |
| Don17 | 47 | F | No | No | 23 | 1,000–10,000 | Frozen | 400 |
| Don18 | 53 | M | No | No | 26 | >10,000 | Fresh | 500 |
| Don19 | 39 | F | Yes | ? | 25 | >10,000 | Frozen | 400 |
| Don21 ** | 37 | M | Yes | No | 34 | 1,000–10,000 | Fresh | 500 |
| Don22 ** | 37 | M | Yes | No | 34 | 1,000–10,000 | Frozen | 400 |
| Don23 | 40 | M | No | No | 29 | 1,000–10,000 | Fresh | 500 |
| Don24 | 48 | M | No | No | 35 | 1,000–10,000 | Fresh | 500 |
| Don25 | 56 | M | Unknown | Unknown | 29 | 1,000–10,000 | Fresh | 500 |
| Don26 | 49 | M | No | No | 30 | 1,000–10,000 | Fresh | 500 |
| Don27 | 31 | M | Unknown | Unknown | 34 | 1,000–10,000 | Fresh | 500 |
| Don29 | 22 | F | Unknown | Unknown | 27 | 1,000–10,000 | Frozen | 400 |
| Don33 *** | 21 | F | No | No | 31 | 1,000–10,000 | Frozen | 400 |
| Don34 | 50 | M | Unknown | Unknown | 39 | 1,000–10,000 | Fresh | 500 |
| Don35 | 55 | M | No | No | 28 | 1,000–10,000 | Fresh | 500 |
| Don36 | 57 | M | No | No | 44 | 1,000–10,000 | Fresh | 500 |
| Don37 | 57 | M | Unknown | Unknown | 34 | >10,000 | Fresh | 500 |
| Don38 | 36 | M | Yes | Yes | 36 | 1,000–10,000 | Fresh | 500 |
| Don39 *** | 21 | F | No | No | 31 | 1,000–10,000 | Frozen | 400 |
| Don40 | 54 | M | No | No | 32 | 1,000–10,000 | Fresh | 500 |

* Donor 10 and Donor 11 are the same

** Don21 and Don22 are the same

*** Donor 33 and Donor 39 are the same

the study, 23 patients met the criteria for Track 2 and 12 patients met the criteria for Track 3. Their subgroup demographics were: Track 2 –median age range 61 years (IQR 21.5); 14 females and 9 males; median BMI 29.9 (IQR 12.06); Track 3 –median age range 50.5 (IQR 11.25); 4 females and 8 males; median BMI 27.3 (IQR 6.68). Four out of the 23 patients

**Table 2. Summary of Donor Plasma Luminex Analyses.**

| Analytes pg/ml | Range | Median | IQR | Mean | SD | % Elevated* |
|---|---|---|---|---|---|---|
| IFNα2 | <8.0–114.55 | 36.17 | 21.04 | 39.92 | 19.4 | 20.0 |
| IL-6 | 0.13–8.72 | 0.64 | 0.698 | 1.01 | 1.48 | 31.4 |
| PCT | 0–2402.57 | 224.72 | 483.13 | 491.03 | 599.7 | 31.4 |
| CRP ug/ml | <0.01–37.26 | 4.12 | 5.81 | 6.99 | 7.29 | 25.7 |
| IG Isotypes ug/ml | | | | | | |
| IgM | 383.60–5695.3 | 832.01 | 439.75 | 1043.31 | 958.55 | 5.7 |
| IgG1 | 1127.2–4754.4 | 2107.78 | 836.49 | 2299.82 | 818.19 | 0.0 |
| IgG2 | 61.1–4039.1 | 1792.28 | 1704.12 | 1859.41 | 1008.65 | 2.9 |
| IgG3 | 80.6–2965.4 | 546.92 | 410.2 | 840.47 | 878.73 | 14.2 |
| IgG4 | 2.6–7539.1 | 347.13 | 2378.29 | 1785.92 | 2416.37 | 34.3 |
| IgA | 372.1–3090.4 | 912 | 705.59 | 1105.89 | 527.7 | 2.9 |

* The % of donors with plasma values elevated above the normal control mean + 2xSD

(17.4%) in Track 2 had sufficient disease progression to necessitate placement on mechanical ventilation after receiving CPT. The number of days from symptom onset until CPT infusion ranged from 2–27 days (which may account for some of the variability seen in cytokine and Ig concentrations in recipient plasma both pre- and post-CPT). Of the 35 patients, 24 developed ARDS (68.6%). Of those patients in Track 2, 52.2% developed ARDS compared with 100% of the patients in Track 3. Furthermore, end organ dysfunction developed in 21.7% of the patients in Track 2 and in 50% of the patients in Track 3. Of the Track 2 patients in this arm of study, 82.6% were discharged alive and in Track 3 58.3% were discharged alive.

## Elevations of cytokine/chemokine concentrations in blood plasma samples of CPT recipients at all time points

We calculated the percentage of recipients with elevated blood plasma cytokine/chemokine concentrations (as defined in Materials & Methods) at each of the three time points (pre-infusion [Day 0], Day 3, and Day 10). The 20 elevated analytes that exhibited a percentage of recipients at any time point that met or exceeded 20% are listed with their descriptive statistics in **Tables 4, 5 and 6** for pre-infusion, Day 3 and Day 10, respectively (individual concentration values are listed in **S7**–**S9 Tables**). The remaining 15 tested analytes exhibiting less than 20% of recipients with elevated concentrations are listed in **S2**–**S4 Tables** for the respective time points.

Those cytokines/chemokines with elevated concentrations in at least 20% of the CPT recipient blood plasma samples are summarized for all time points in **Fig 2A**. Analytes associated with an acute inflammatory response such as IL-6, IL-8, IP-10, as well as CRP, were elevated in over 90% of the patients at Pre-Infusion. IL-7, MCP-1, and MIP-1β were also initially elevated in between 50–90% of the patient samples. In addition, while the proportion of patient samples with elevated EGF, IL-1RA, and NGAL was initially low, the percentage of patients' blood plasma samples with elevated concentrations increased to over 20% in these three cytokines by Day 10 (**Fig 2A**).

By treating cytokines/chemokines as binary variables (elevated/non-elevated as defined in Materials & Methods), the change in the percentage of patients with elevated blood plasma cytokine levels was tested by the Cochran-Armitage test. For the Track 2 cohort, the percentage of patients with elevated levels of EGF (p = 0.005), IL-1RA (p<0.001), VEGF (p = 0.005), and NGAL (p<0.001) were found to increase significantly over the course of 10 days (**Fig 2B**).

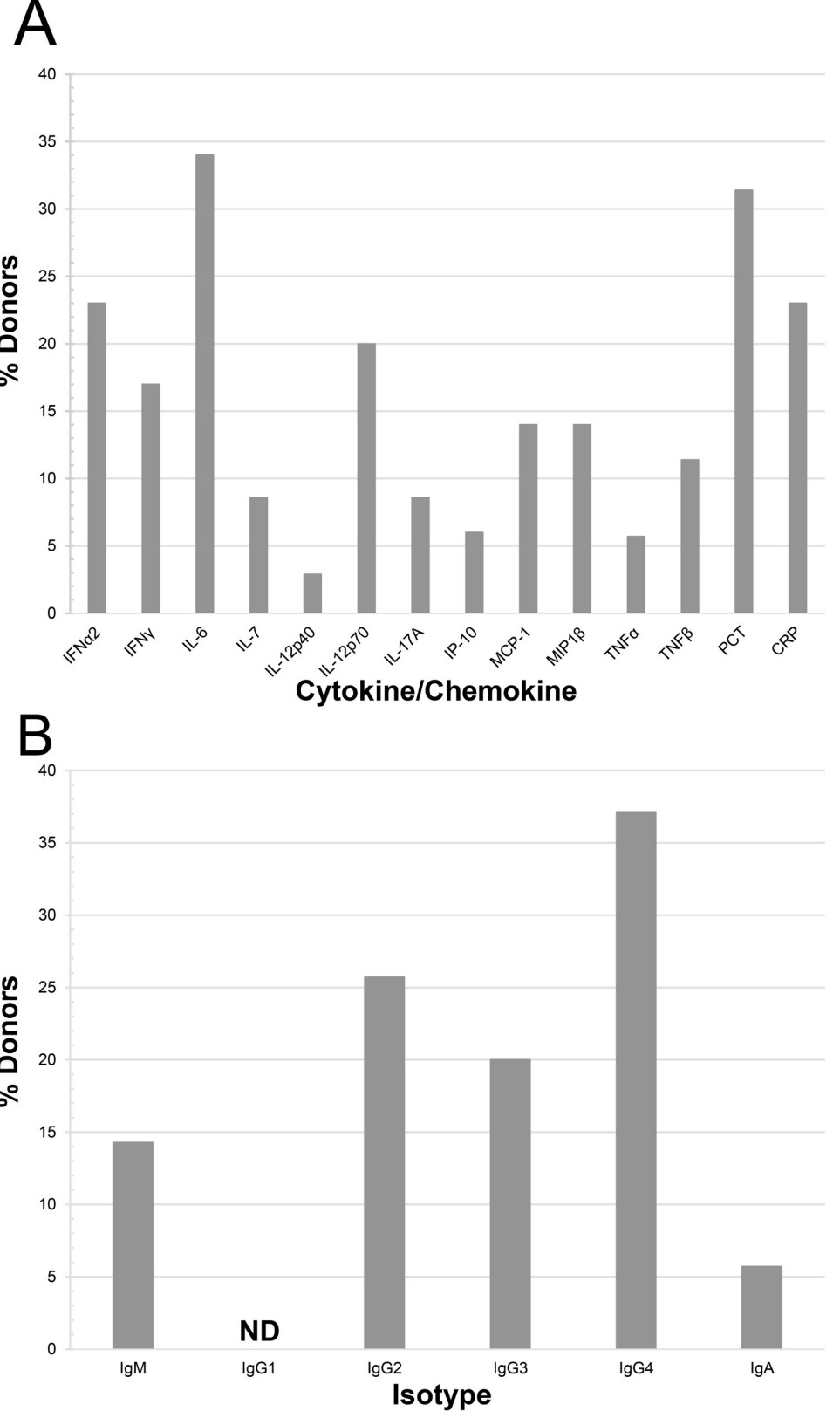

**Fig 1. Composition of donor plasma samples.** Donor plasma samples (n = 35) were analyzed for concentration of inflammatory cytokines/chemokines and Ig isotypes using multiplex immunoassays via the Luminex platform and

analyzed using Belysa software. Alternatively, ELISA analysis was performed for measurement of PCT concentration. Samples were compared to healthy control plasma and were marked as elevated if the concentration was greater than the mean of the control samples plus two times the standard deviation (n = 12–16). (A) The percentage of donor samples with elevated values was calculated for each analyte, 14 of which exhibited any level of elevation. (B) The percentage of donors with elevation in the concentration of Ig Isotypes. ND, none detected.

Conversely, there was a significant decrease in the percentage of patients' blood plasma samples with elevated levels over time of IFN-α2 (p<0.001), IFN-γ (p<0.001), IL-3 (p<0.001), IL-7 (p = 0.0012), IL-12p40 (p = 0.003), IL-12p70 (p = 0.006), IL-17A (p = 0.028), TNF-β (p = 0.043), and CRP (p = 0.003). For Track 3, there was a significant increase in the percentage of patients' blood plasma samples with elevations in IL-1RA (p<0.001) and NGAL (p<0.001), while there was a significant decrease in the percentage of patients' samples with elevated RANTES (p = 0.005) over the course of 10 days (**Fig 2C**).

## Changes in the mean blood plasma cytokine concentrations in CPT recipients over time

The means of the CPT recipients' blood plasma cytokine/chemokine concentrations over time are depicted in **Fig 3**, divided into three sets with similar ranges of values. Each set includes a comparison for the total patient cohort (**Fig 3A, 3D and 3G**), and separated into Track 2 (**Fig 3B, 3E and 3H**) or Track 3 (**Fig 3C, 3F and 3I**) patients. For Track 2 patients, the mean concentrations of several cytokines were found to significantly decrease over time. These included IFN-α2 (Day 0 to Day 10, p<0.001; Day 3 to Day 10, p = 0.038), IFN-γ (Day 0 to Day 3; Day 0 to Day 10, p<0.001), IL-3 (Day 0 to Day 3, p = 0.020; Day 0 to Day 10, p<0.001), IL-12p40 (Day 0 to Day 10, p = 0.020), IL-12p70 (Day 0 to Day 10, p = 0.006), IL-17A (Day 0 to Day 10, p = 0.036) (**Fig 3B**), CRP (Day 0 to Day 10, p = 0.022; Day 3 to Day 10, p = 0.21) (**Fig 3E**) and RANTES (Day 0 to Day 10, p = 0.043; Day 3 to Day 10, p = 0.010) (**Fig 3H**). Others were found to significantly increase over time for Track 2 patients—EGF (Day 0 to Day 10, p = 0.019; **Fig 3B**), VEGF (Day 0 to Day 10, p = 0.019; **Fig 3E**), and NGAL (Day 0 to Day 10; Day 3 to Day 10, p,0.001; **Fig 3H**). In comparison, for Track 3 patients the mean blood plasma concentrations of only a few cytokines were found to change significantly over time. The mean concentrations of IL-1RA (Day 0 to Day 10, p = 0.014; **Fig 3F**), TNF-α (Day 0 to Day 10, p = 0.05; **Fig 3C**), and NGAL (Day 0 to Day 10, p = 0.021; **Fig 3I**) increased over the course of 10 days, while the concentration of RANTES (Day 0 to Day 3, p = 0.028; Day 0 to Day 10, p = 0.007; **Fig 3I**) significantly decreased in the same time frame. Interestingly, the mean TNF-α concentration peaked at Day 3 and decreased about half on Day 10 but with a mean concentration still significantly above the pre-infusion levels (**Fig 3C**). Also, for Track 3, five other analytes trended to peak at Day 3 and then decline, albeit without statistical significance, including IL-8, MIP-1β (**Fig 3C**), IL-6, CRP (**Fig 3F**) and MCP-1 (**Fig 3I**).

Furthermore, we examined the differences in the mean blood plasma cytokine/chemokine levels in patients in Track 2 compared to those in Track 3. The mean concentrations of several cytokines were found to be significantly higher in the Pre-Infusion samples of patients in Track 2 compared to Track 3. These included IFN-α2 (52.49 pg/ml vs 18.63 pg/ml, p<0.001), IFN-γ (53.97 pg/ml vs 15.61 pg/ml, p = 0.005), IL-1RA (15.74 pg/ml vs 8.36 pg/ml, p = 0.038), IL-3 (2.60 pg/ml vs 1.17 pg/ml, p<0.001), IL-12p40 (60.21 pg/ml vs 12.96 pg/ml, p = 0.002), IL-12p70 (5.77 pg/ml vs 2.55 pg/ml, p = 0.004), IL-17A (13.03 pg/ml vs 4.67 pg/ml, p = 0.013), and TNF-α (30.90 pg/ml vs 14.98 pg/ml, p = 0.001) (**Table 4**). Although the difference in mean concentration of IL-6 at the Day 3 and Day 10 time points is striking in Track 3 as compared to patients in Track 2 (**Fig 3E and 3F**), these differences were not found to be statistically

**Table 3. Recipient Characteristics.**

| | Track 2 N = 23 | Track 3 N = 12 | P Value± |
|---|---|---|---|
| **Age*** | 61 (21.5) | 50.5 (11.25) | 0.054 |
| **Sex** | | | 0.16 |
| Male | 39.1% | 66.7% | |
| Female | 60.9% | 33.3% | |
| **BMI*** | 29.9 (12.06) | 27.3 (6.68) | 0.96 |
| **Race** | | | 0.84 |
| Caucasian | 17.4% | 25.0% | |
| African American | 4.3% | 8.3% | |
| Hispanic | 60.9% | 58.3% | |
| Asian | 17.4% | 8.3% | |
| **Days from symptom onset to treatment*** | 10 (10.5) | 16 (9.5) | 0.0063 |
| **Day 10 COVID swab PCR^** | | | 0.6 |
| Positive | 54.5% | 10.0% | |
| Negative | 45.5% | 30.0% | |
| Not Determined | | 60.0% | |
| **Day 30 survival status** | | | 0.015 |
| Alive | 82.6% | 58.3% | |
| Fatal | 17.4% | 41.7% | |
| **Intubated after treatment** | | | |
| Yes | 17.4% | NA | |
| No | 82.6% | NA | |
| **Days from infusion to discharge or death*** | 8 (9) | 10 (7) | 0.062 |
| **Developed ARDS** | 52.2% | 100% | 0.0055 |
| **Developed End Organ Dysfunction** | 21.7% | 50.0% | 0.13 |
| **Concomitant Treatment** | | | |
| Hydroxychloroquine | 69.6% | 91.7% | 0.22 |
| Azithromycin | 52.2% | 75% | 0.28 |
| Doxycycline or other antibiotic | 56.5% | 50% | 0.74 |
| Steroids | 52.2% | 100% | 0.0055 |
| Tociluzumab | 17.4% | 58.3% | 0.022 |
| Remdesivir | 4.3% | 8.3% | 1 |
| **Comorbidities** | | | |
| Hypertension | 47.8% | 33.3% | 0.49 |
| Diabetes | 34.8% | 50.0% | 0.48 |
| Heart Failure or CAD | 4.3% | 0% | 1 |
| History of Smoking | 17.4% | 16.7% | 0.51 |
| Lung disease or Asthma | 26.1% | 8.3% | 0.38 |
| Immunocompromised | 30.4% | 8.3% | 0.22 |
| Active Cancer | 26.1% | 8.3% | 0.38 |

*Median (IQR)

^Track 2, n = 22; Track 3, n = 10

±Continuous variables were analyzed using T-test; Categorical values were analyzed using Fisher's exact test

significant (p = 0.183 and p = 0.185, respectively), as the elevated means could be attributed to high levels in only a few patients within the Track 3 cohort (REC18 and REC34 at Day 3, 7857.43 pg/ml and 9496.55 pg/ml, respectively, **S8 Table**; REC 18, REC19, and REC21 at Day 10, 3377.42 pg/ml, 1101.30 pg/ml, and 2746.72 pg/ml, respectively, **S9 Table**). Similarly, IL-8

**Table 4. Summary of Pre-Infusion Recipient Cytokine Levels of Interest.**

| Analyte pg/ml | Range | Median | IQR | Mean | SD | % Elevated* | Analyte pg/ml | Range | Median | IQR | Mean | SD | % Elevated* | P Value^ |
|---|---|---|---|---|---|---|---|---|---|---|---|---|---|---|
| | | | Track 2 | | | | | | | | Track 3 | | | |
| EGF | <3.2–80.40 | 13.84 | 9.07 | 16.68 | 18.17 | 8.7 | EGF | <3.2–62.91 | 13.59 | 16.04 | 18.73 | 16.79 | 16.7 | 0.370 |
| IFNα2 | 1.39–124.05 | 51.53 | 56.82 | 52.49 | 37.82 | 43.5 | IFNα2 | <8.0–50.41 | 10.97 | 24.64 | 18.63 | 15.73 | 0.0 | 0.000 |
| IFNγ | <1.28–193.56 | 44.22 | 48.78 | 53.97 | 55.63 | 65.2 | IFNγ | <1.28–92.90 | 5.65 | 11.90 | 15.61 | 26.50 | 16.7 | 0.005 |
| IL-1RA | 5.41–65.68 | 12.58 | 11.39 | 15.74 | 13.26 | 4.4 | IL-1RA | 0.91–25.41 | 5.86 | 6.38 | 8.36 | 6.95 | 0.0 | 0.019 |
| IL-3 | <1.28–6.63 | 2.04 | 1.92 | 2.60 | 1.77 | 60.9 | IL-3 | <1.28–1.19 | 1.28 | 0.10 | 1.17 | 0.22 | 0.0 | 0.000 |
| IL-6 | 2.34–65.13 | 11.43 | 27.03 | 21.50 | 20.85 | 95.7 | IL-6 | 1.49–117.47 | 34.16 | 52.55 | 39.37 | 36.35 | 100.0 | 0.068 |
| IL-7 | <0.64–42.90 | 2.52 | 3.74 | 5.83 | 9.15 | 82.6 | IL-7 | <0.64–17.02 | 1.46 | 3.35 | 4.00 | 5.66 | 75.0 | 0.240 |
| IL-8 (CXCL8) | 2.49–48.94 | 6.67 | 6.54 | 11.41 | 13.14 | 100.0 | IL-8 (CXCL8) | 2.12–43.16 | 12.20 | 20.83 | 16.49 | 13.17 | 91.7 | 0.140 |
| IL-12p40 | 5.99–321.69 | 42.08 | 35.04 | 60.21 | 67.78 | 47.8 | IL-12p40 | <6.4–35.24 | 7.70 | 10.59 | 12.96 | 10.95 | 0.0 | 0.002 |
| IL-12p70 | 0.12–17.86 | 4.49 | 6.19 | 5.77 | 5.09 | 52.2 | IL-12p70 | <3.2–4.87 | 3.20 | 1.74 | 2.55 | 1.36 | 8.3 | 0.004 |
| IL-17A | 1.05–75.73 | 11.39 | 15.31 | 13.03 | 16.04 | 26.1 | IL-17A | 1.05–14.91 | 2.09 | 6.08 | 4.67 | 4.55 | 8.3 | 0.013 |
| IP-10 (CXCL10) | 55.74–40124.31 | 1390.40 | 1604.04 | 5408.34 | 11256.48 | 95.7 | IP-10 (CXCL10) | 134.54–35606.28 | 613.59 | 1101.74 | 3693.77 | 10069.47 | 91.7 | 0.330 |
| MCP-1 (CCL2) | 164.02–6568.98 | 508.60 | 598.99 | 890.59 | 1313.74 | 65.2 | MCP-1 (CCL2) | 140.87–3454.87 | 490.67 | 1086.64 | 995.78 | 1000.51 | 75.0 | 0.400 |
| MIP-1β (CCL4) | 13.35–176.16 | 38.42 | 16.23 | 42.29 | 31.61 | 60.9 | MIP-1β (CCL4) | 12.38–92.26 | 31.52 | 19.84 | 37.28 | 21.66 | 50.0 | 0.290 |
| RANTES (CCL5) | 147.11–18767.63 | 3851.95 | 1629.52 | 5994.50 | 5882.80 | 26.1 | RANTES (CCL5) | 2410.40–13120.73 | 5365.12 | 7852.69 | 6833.84 | 4052.37 | 66.7 | 0.310 |
| TNFα | 2.65–89.44 | 28.22 | 18.35 | 30.90 | 19.78 | 39.1 | TNFα | 4.53–29.38 | 13.18 | 7.26 | 14.98 | 6.53 | 0.0 | 0.001 |
| TNFβ | <1.6–16.60 | 3.28 | 8.15 | 6.38 | 6.50 | 34.8 | TNFβ | <1.6–16.98 | 1.60 | 1.84 | 3.58 | 4.43 | 8.3 | 0.072 |
| VEGF | <2.56–272.71 | 33.16 | 123.60 | 72.97 | 93.08 | 34.8 | VEGF | 1.38–603.16 | 98.62 | 72.49 | 136.89 | 171.90 | 66.7 | 0.140 |
| CRP ug/ml | 26.13–1033.63 | 256.18 | 337.65 | 304.36 | 238.76 | 100.0 | CRP ug/ml | 8.33–1358.15 | 139.30 | 245.29 | 274.75 | 390.57 | 100.0 | 0.410 |
| NGAL ng/ml | 23.7–586.3 | 233.80 | 200.57 | 252.95 | 141.70 | 13.0 | NGAL ng/ml | 108.5–2226.28 | 284.23 | 171.30 | 468.13 | 583.97 | 25.0 | 0.120 |

* The % of recipients with plasma values elevated above the normal control mean + 2xSD

^T-test was used to compare Means of Track 2 and Track 3; Statistically significant p values are highlighted in gray

SD, Standard Deviation of the Mean; IQR, Interquartile Range

concentrations were also extremely elevated in REC18 (383.11 pg/ml) and REC34 (9496.55 pg/ml) at Day 3 (S8 Table) accounting for the increased mean concentration of this cytokine in the Track 3 cohort (Fig 3B and 3C).

Of the nine analytes that at some time point were elevated in more than 70% of the CPT recipients' blood plasma samples in the combined cohort (Fig 2A), seven of them, i.e. IL-6, IL-7, IL-8, IP-10, MCP-1, VEGF and NGAL are plotted in Fig 4 to follow individual levels,

**Table 5. Summary of Day 3 Recipient Cytokine Levels of Interest.**

| Analyte pg/ml | Range | Median | IQR | Mean | SD | % Elevated* | Analyte pg/ml | Range | Median | IQR | Mean | SD | % Elevated* | P Value^ |
|---|---|---|---|---|---|---|---|---|---|---|---|---|---|---|
| | | **Track 2** | | | | | | | **Track 3** | | | | | |
| EGF | 3.17–133.43 | 9.31 | 29.82 | 25.59 | 32.06 | 26.09 | EGF | <3.2–59.59 | 19.65 | 24.02 | 21.63 | 17.45 | 25.00 | 0.320 |
| IFNα2 | 1.67–151.11 | 27.41 | 23.70 | 37.60 | 35.74 | 21.74 | IFNα2 | <8–59.24 | 17.21 | 11.27 | 22.63 | 16.98 | 16.67 | 0.051 |
| IFNγ | <1.28–53.36 | 4.73 | 6.61 | 8.86 | 12.29 | 8.70 | IFNγ | <1.28–14.73 | 2.28 | 8.53 | 5.13 | 4.83 | 0.00 | 0.105 |
| IL-1RA | 3.05–188.26 | 9.14 | 10.21 | 21.31 | 39.38 | 8.70 | IL-1RA | 9.25–337.25 | 22.43 | 60.79 | 74.28 | 113.05 | 33.33 | 0.070 |
| IL-3 | 0.15–6.46 | 1.28 | 0.48 | 1.38 | 1.30 | 13.04 | IL-3 | 0.72–1.48 | 1.28 | 0.00 | 1.25 | 0.18 | 8.33 | 0.296 |
| IL-6 | 0.58–1276.07 | 7.44 | 33.10 | 101.63 | 285.37 | 95.65 | IL-6 | 4.44–9496.55 | 20.59 | 116.98 | 1488.69 | 3377.16 | 100.00 | 0.092 |
| IL-7 | 0.13–49.12 | 1.88 | 3.00 | 4.83 | 10.38 | 69.56 | IL-7 | 0.10–4.20 | 0.64 | 0.71 | 1.10 | 1.10 | 33.33 | 0.051 |
| IL-8 (CXCL8) | 2.04–87.94 | 7.53 | 5.02 | 15.40 | 21.90 | 95.65 | IL-8 (CXCL8) | 1.14–2260.01 | 11.52 | 11.22 | 228.73 | 648.64 | 91.67 | 0.140 |
| IL-12p40 | 3.42–148.86 | 25.15 | 29.77 | 35.96 | 34.95 | 21.74 | IL-12p40 | 4.50–25.15 | 11.57 | 11.41 | 12.99 | 7.91 | 0.00 | 0.003 |
| IL-12p70 | 0.70–25.68 | 3.20 | 2.02 | 4.98 | 6.10 | 30.43 | IL-12p70 | 0.41–6.35 | 3.20 | 0.62 | 2.88 | 1.72 | 16.67 | 0.067 |
| IL-17A | <1.28–59.54 | 5.99 | 7.32 | 10.03 | 13.99 | 13.04 | IL-17A | 0.15–15.87 | 2.88 | 4.56 | 4.84 | 4.85 | 8.33 | 0.060 |
| IP-10 (CXCL10) | 44.15–35687.71 | 436.18 | 954.14 | 2972.42 | 7918.82 | 91.30 | IP-10 (CXCL10) | 33.33–17302.90 | 725.81 | 658.63 | 2199.46 | 4819.52 | 91.67 | 0.360 |
| MCP-1 (CCL2) | 134.73–41880.15 | 477.42 | 361.56 | 2719.01 | 8664.70 | 73.91 | MCP-1 (CCL2) | 152.08–6642.23 | 666.43 | 780.28 | 1484.34 | 2126.76 | 66.67 | 0.260 |
| MIP-1β (CCL4) | 22.13–184.66 | 33.75 | 10.94 | 41.91 | 32.58 | 52.17 | MIP-1β (CCL4) | 19.74–488.89 | 46.35 | 29.89 | 110.88 | 167.48 | 66.67 | 0.090 |
| RANTES (CCL5) | 912.62–7083.55 | 4435.97 | 2194.08 | 4469.85 | 1611.57 | 47.82 | RANTES (CCL5) | 1967.0–5025.05 | 4060.78 | 1499.68 | 3842.64 | 995.29 | 25.00 | 0.080 |
| TNFα | 8.03–99/03 | 25.36 | 13.70 | 33.88 | 24.32 | 30.43 | TNFα | 6.76–564.52 | 21.88 | 18.77 | 89.18 | 166.80 | 33.33 | 0.139 |
| TNFβ | 0.70–24.53 | 2.03 | 5.89 | 5.69 | 6.78 | 26.09 | TNFβ | <1.6–23.70 | 3.05 | 7.55 | 7.16 | 8.07 | 25.00 | 0.298 |
| VEGF | <2.56–464.61 | 30.61 | 134.69 | 98.37 | 128.97 | 39.13 | VEGF | <2.56–322.02 | 54.70 | 118.32 | 96.87 | 109.20 | 50.00 | 0.490 |
| CRP ug/ml | 4.97–1589.72 | 249.45 | 334.06 | 362.55 | 395.50 | 95.65 | CRP ug/ml | 0.56–2216.15 | 212.90 | 601.06 | 480.52 | 648.39 | 91.67 | 0.290 |
| NGAL ng/ml | 69.34–849.16 | 243.69 | 213.77 | 319.71 | 255.27 | 13.04 | NGAL ng/ml | 184.56–888.91 | 332.26 | 263.95 | 403.46 | 583.97 | 33.33 | 0.170 |

* The % of recipients with plasma values elevated above the normal control mean + 2xSD

^T-test was used to compare Means of Track 2 and Track 3; Statistically significant p values are highlighted in gray

SD, Standard Deviation of the Mean; IQR, Interquartile Range

separated by Track. The remaining two analytes, CRP and MIP-1β are discussed in the next section. As can be seen, some individuals in each Track exhibited unusually high levels of the relevant cytokine/chemokine in comparison to the rest of the group. For example, with IL-6, five out of 23 recipients in Track 2 expressed concentration levels above 150 pg/ml, whereas three out of 12 recipients in Track 3 were above that level (**Fig 4A**). Some of the cytokines/chemokines reached high levels early on and then dropped by Day 10, as in the case of IP-10 (**Fig 4D**). Other analytes reached their high points at Day 10 in both Tracks, such as with NGAL (**Fig 4G**).

**Table 6. Summary of Day 10 Recipient Cytokine Levels of Interest.**

| Analyte pg/ml | Track 2 | | | | | | Analyte pg/ml | Track 3 | | | | | | P Value^ |
| | Range | Median | IQR | Mean | SD | % Elevated* | | Range | Median | IQR | Mean | SD | % Elevated* | |
|---|---|---|---|---|---|---|---|---|---|---|---|---|---|---|
| EGF | 2.18–133.09 | 26.98 | 51.66 | 37.86 | 36.01 | 45.5 | EGF | <3.2–51.17 | 19.43 | 10.07 | 21.58 | 12.77 | 10.0 | 0.035 |
| IFNα2 | 4.83–60.70 | 12.49 | 22.95 | 19.19 | 17.09 | 9.1 | IFNα2 | 3.12–42.98 | 8.00 | 16.43 | 15.33 | 15.28 | 0.0 | 0.270 |
| IFNγ | <1.28–24.11 | 1.28 | 7.58 | 6.28 | 6.77 | 0.0 | IFNγ | <1.28–87.87 | 10.54 | 18.80 | 23.87 | 33.54 | 20.0 | 0.067 |
| IL-1RA | 4.41–3985.12 | 52.94 | 98.48 | 248.89 | 838.14 | 54.6 | IL-1RA | 28.06–308.19 | 66.91 | 50.22 | 88.29 | 82.57 | 70.0 | 0.190 |
| IL-3 | 0.10–2.03 | 1.28 | 0.00 | 1.10 | 0.48 | 4.6 | IL-3 | 0.37–1.28 | 1.28 | 0.46 | 1.05 | 0.38 | 0.0 | 0.380 |
| IL-6 | 0.32–1501.16 | 3.22 | 9.11 | 156.06 | 384.40 | 95.5 | IL-6 | 1.15–3377.42 | 57.86 | 832.13 | 743.11 | 1275.98 | 100.0 | 0.093 |
| IL-7 | 0.18–141.35 | 0.64 | 3.54 | 10.65 | 31.60 | 36.4 | IL-7 | <0.64–4.92 | 0.64 | 0.78 | 1.32 | 1.37 | 30.0 | 0.091 |
| IL-8 (CXCL8) | 2.0–158.35 | 5.82 | 7.55 | 18.41 | 35.31 | 95.5 | IL-8 (CXCL8) | 4.72–598.66 | 10.43 | 8.39 | 71.45 | 185.55 | 100.0 | 0.200 |
| IL-12p40 | 1.37–102.75 | 16.34 | 16.26 | 22.76 | 24.01 | 9.1 | IL-12p40 | 4.18–32.85 | 14.37 | 9.32 | 15.53 | 0.09 | 0.0 | 0.120 |
| IL-12p70 | 0.75–5.45 | 3.20 | 0.00 | 3.13 | 1.09 | 13.6 | IL-12p70 | 1.2–2.86 | 3.20 | 0.60 | 2.79 | 0.67 | 0.0 | 0.150 |
| IL-17A | 0.27–20.44 | 2.57 | 6.11 | 5.09 | 5.83 | 9.1 | IL-17A | 0.27–14.70 | 3.60 | 8.13 | 5.48 | 5.49 | 0.0 | 0.430 |
| IP-10 (CXCL10) | 84.38–24502.52 | 393.99 | 305.80 | 1863.33 | 5200.96 | 81.8 | IP-10 (CXCL10) | 223.02–2569.14 | 492.86 | 308.44 | 664.31 | 698.85 | 100.0 | 0.150 |
| MCP-1 (CCL2) | 158.37–2893.75 | 418.43 | 495.94 | 695.71 | 745.60 | 54.6 | MCP-1 (CCL2) | 182.03–3759.13 | 479.75 | 541.03 | 1064.89 | 1258.67 | 90.0 | 0.200 |
| MIP-1β (CCL4) | 22.83–218.53 | 39.91 | 13.87 | 54.87 | 46.93 | 81.8 | MIP-1β (CCL4) | 13.02–164.13 | 34.24 | 27.72 | 50.95 | 43.11 | 70.0 | 0.410 |
| RANTES (CCL5) | 1741.10–6677.48 | 2967.81 | 1648.37 | 3304.97 | 1257.30 | 9.1 | RANTES (CCL5) | 1788.05–5755.71 | 2752.26 | 782.41 | 2949.10 | 1112.64 | 10.0 | 0.215 |
| TNFα | 1.80–321.88 | 19.72 | 16.37 | 40.35 | 69.56 | 22.7 | TNFα | 14.44–145.46 | 26.09 | 42.83 | 46.81 | 45.19 | 30.0 | 0.380 |
| TNFβ | 2.20–11.00 | 3.06 | 1.98 | 4.02 | 2.47 | 9.1 | TNFβ | <2.56–14.35 | 4.98 | 7.56 | 7.00 | 4.80 | 40.0 | 0.045 |
| VEGF | <2.56–853.21 | 139.14 | 138.94 | 208.60 | 201.00 | 77.3 | VEGF | 8.02–664.70 | 151.71 | 273.96 | 249.70 | 226.80 | 90.0 | 0.320 |
| CRP ug/ml | 1.24–1357.70 | 20.21 | 80.35 | 116.05 | 286.54 | 72.7 | CRP ug/ml | 5.04–672.39 | 72.06 | 122.16 | 149.87 | 210.47 | 80.0 | 0.360 |
| NGAL ng/ml | 314.65–4192.62 | 1463.85 | 1712.12 | 1602.48 | 1147.21 | 86.4 | NGAL ng/ml | 575.42–6187.99 | 1175.22 | 1554.81 | 2073.92 | 1803.37 | 100.0 | 0.230 |

* The % of recipients with plasma values elevated above the normal control mean + 2xSD

^T-test was used to compare Means of Track 2 and Track 3; Statistically significant p values are highlighted in gray

SD, Standard Deviation of the Mean; IQR, Interquartile Range

## Association of changes in blood plasma cytokine/chemokine levels with clinical endpoints

Logistic regression analysis was performed to look for associations between changes in recipient blood plasma cytokine/chemokine concentrations and defined clinical endpoints (Table 7). For patients in Track 2 the clinical endpoint was defined as progression to mechanical ventilation, while for those in Track 3 the defined clinical endpoint was incidence of mortality by Day 30 post-infusion. Analysis of all analytes revealed that only three had statistically significant correlations with Track 2 patient progression to intubation, involving four

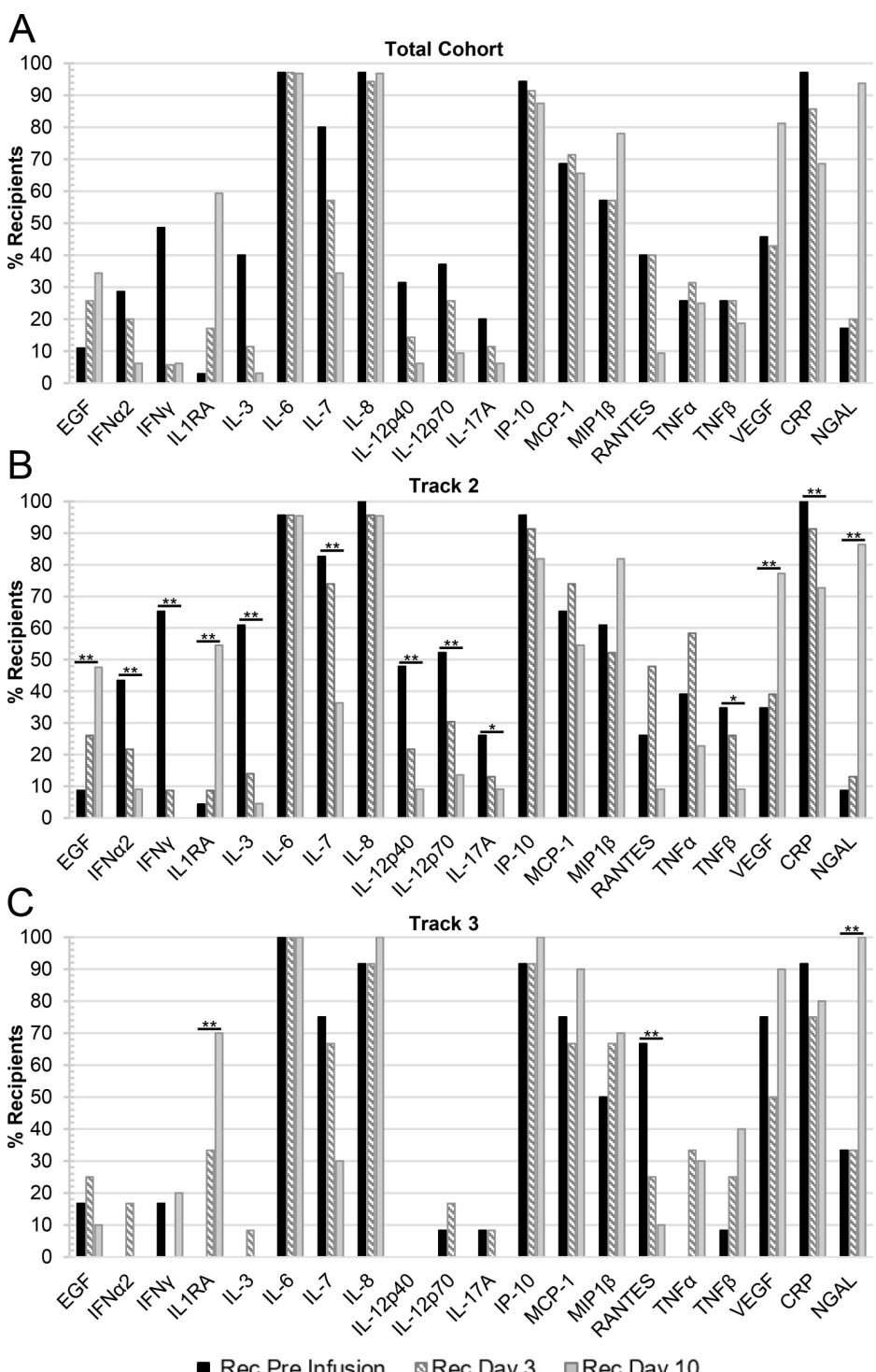

**Fig 2. The percentage of CPT recipients with elevated cytokine/chemokine concentrations in their plasma over all time points.** CPT recipient plasma samples were compared to healthy control plasma and were marked as elevated if the concentration was greater than the mean of the control samples plus two times the standard deviation (n = 12–16). Included in these summary figures are the 20 cytokines/chemokines that were elevated in at least 20% of the recipients at any of the time points examined and grouped based on either: (A) the total cohort of recipients (Pre-infusion, Day 3, n = 35; or Day 10, n = 32); (B) recipients in Track 2 (Pre-infusion, Day 3, n = 23; or Day 10, n = 22); or (C) recipients in Track 3 (Pre-infusion, Day 3, n = 12; or Day 10, n = 10). Statistically significant changes in mean values between time points in the Track 2 and Track 3 cohorts are denoted by a bar and single asterisk (*) for 0.01≤p≤0.05 or a double asterisk (**) for p<0.01.

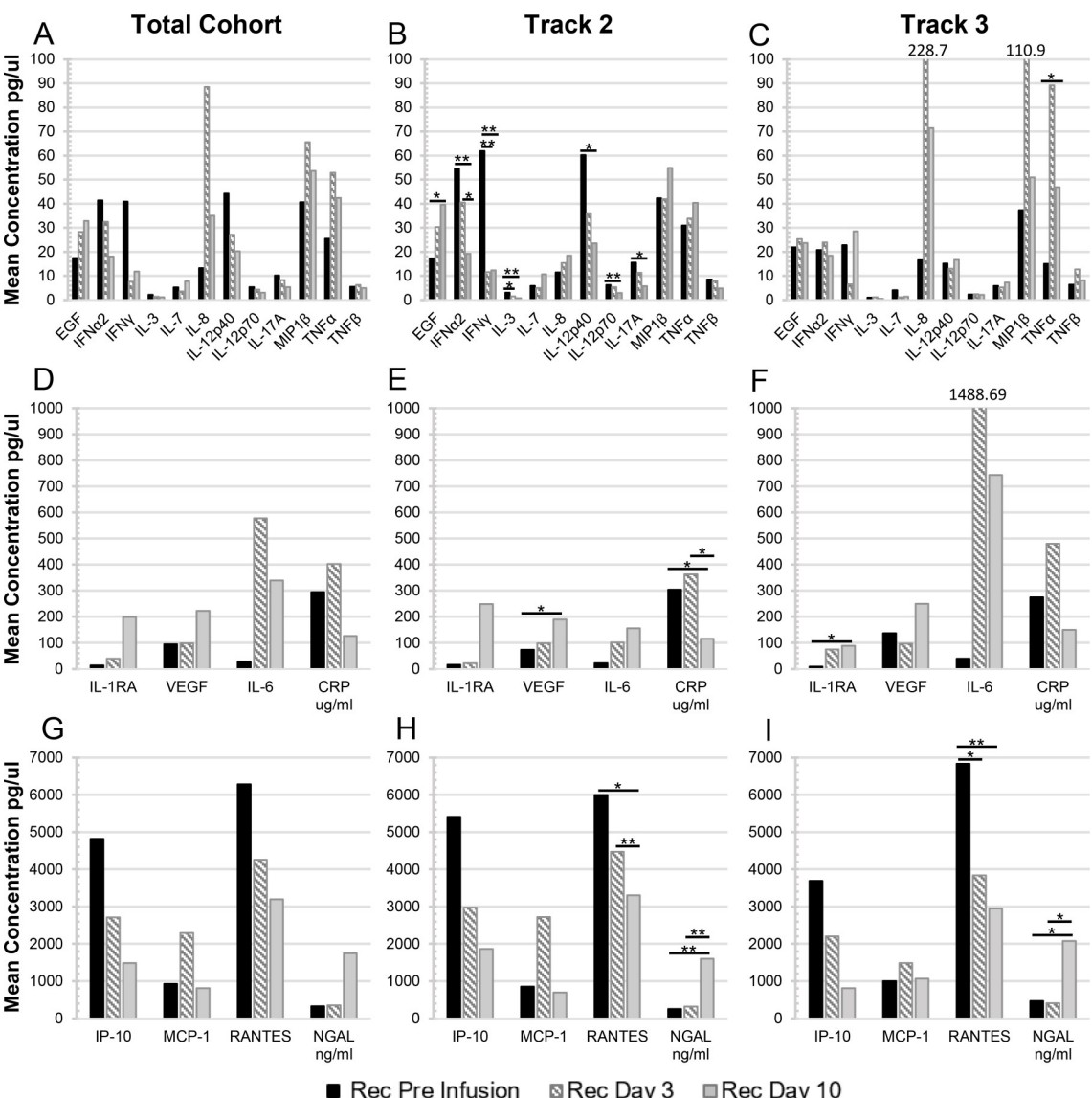

**Fig 3. The mean concentrations of elevated cytokines/chemokines in the plasma of CPT recipients over all time points.** These figures summarize the mean concentration of 20 cytokines/chemokines that were elevated in at least 20% of the recipients at any of the time points (Fig 2), and grouped based on similar ranges of concentration values: (A-C) 0–100; (D-F) 0–1000; and (G-I) 0–7000 units/ml. The mean concentration values are also shown separated for: (A, D, G) the total cohort of recipients (Pre-infusion, Day 3, n = 35; or Day 10, n = 32); (B, E, H) recipients in Track 2 (Pre-infusion, Day 3, n = 23; or Day 10, n = 22); or (C, F, I) recipients in Track 3 (Pre-infusion, Day 3, n = 12; or Day 10, n = 10). Statistically significant changes in mean values between time points in the Track 2 and Track 3 cohorts are denoted by (*) for 0.01≤p≤0.05 or (**) for p<0.01.

recipients (REC08, REC22, REC27, and REC37). Increasing CRP levels from Day 0 to Day 3 correlated with Track 2 patients requiring intubation (p = 0.028; Odds Ratio = 1.0062; **Fig 5A**). Similarly, elevations of MIP-1β from Day 3 to Day 10 was associated with intubation (p = 0.049; Odds Ratio = 1.1146; **Fig 5B**). Interestingly, although MIP-1α levels were not elevated in a significant proportion of patients' plasma samples at any time point, elevations in this cytokine from Day 3 to Day 10 were also found to correlate with progression to intubation (p = 0.037; Odds Ratio = 1.1154; **Fig 5C**). No cytokine/chemokine elevations were found to be significantly associated with day 30 mortality for patients in Track 3. In addition, Chi-square

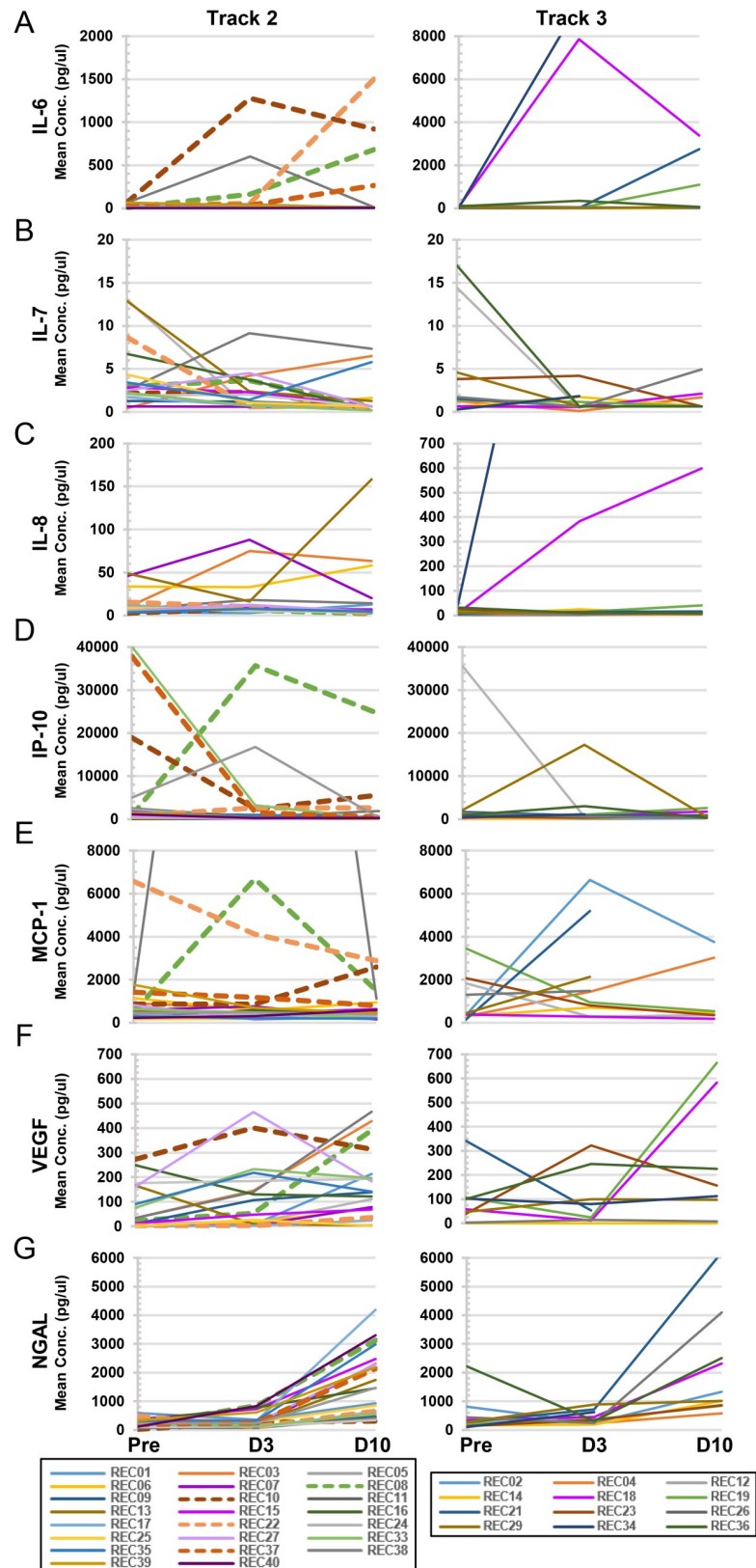

**Fig 4. Individual concentration plots for highly elevated cytokines/chemokines.** Analytes that were elevated in at least 70% of recipients at any time point were plotted for individual recipients in either Track 2 (left-hand panels; Pre-

infusion, Day 3, n = 23; or Day 10, n = 22) or Track 3 (right-hand panels; Pre-infusion, Day 3, n = 12; or Day 10, n = 10). These included: (A) IL-6; (B) IL-7; (C) IL-8; (D) IP-10; (E) MCP-1; (F) VEGF; and (G) NGAL. Dashed lines indicate Track 2 patients who progressed to intubation post-infusion.

tests were used to probe for associations between changes in cytokine/chemokine levels and progression to ARDS or end organ dysfunction. No significant correlations were found.

## Antibody titers and Ig isotype analysis of blood plasma from CPT recipients

Recipient titers of IgG and IgM SARS-CoV2 RBD neutralizing antibodies were measured pre-infusion and on Day 3 and Day 10 post-infusion (Table 8). Eleven of the 35 patients (31.4%) analyzed in this arm of the study were non-immune or minimally immune as defined by neutralizing IgG titers < 1:500. An increase in neutralizing IgG titers was observed in the blood plasma of all eleven patients post-infusion. We also measured plasma Ig isotype concentrations in CPT recipients blood plasma over time, as detailed in Table 9 and which are summarized and graphically separated by Track in Fig 6. As with the cytokine/chemokine analyses, Ig concentrations were defined as elevated if they were above the normal control means plus 2xSD. Both IgG3 and IgG4 were found be elevated in the blood plasma of a significant percentage of patients by Day 10 post-infusion (37.5% and 34.4% respectively, Table 9), particularly evident in Track 2 patients (Fig 6D and 6E). T-test comparisons of mean concentrations at the different time points found that for Track 2 patients, mean concentrations of IgM and IgG3 increased significantly from Pre-Infusion to Day 10 (p = 0.033 and p = 0.004, respectively) and from Day 3 to Day 10 (p = 0.018 for both); IgG1 significantly increased at all time points (Day 0 to Day 3, p = 0.003; Day 0 to Day 10, p<0.001, Day 3 to Day 10, p = 0.001); IgG2 significantly increased from Pre-Infusion to Day 3 (p = 0.035) and from Day 3 to Day 10 (p = 0.026); and IgA significantly increased from Day 3 to Day 10 (p = 0.041). On the other hand, no significant changes in mean Ig isotype concentrations were found in the Track 3 cohort. Finally, comparison of mean concentrations of Ig isotypes between Track 2 and Track 3 revealed a statistically significant difference in the level of IgG4 at the Pre-Infusion stage (1098.0 ug/ml vs. 325.3 ug/ml, p = 0.028) and Day 10 post-infusion (1883.73 ug/ml vs 657.25 ug/ml, p = 0.045) (Table 9).

## Comparison of changes in plasma cytokine concentrations over time in CPT patients compared to nonCPT patients

We sought to determine if changes in cytokine concentrations over time in CPT patients were significantly different than those patients who did not receive CPT (nonCPT). Cryopreserved plasma samples from hospitalized COVID-19 patients from Track 2 and Track 3 were obtained from our institutional biorepository. The nonCPT samples were chosen to correlate with a similar time from symptom onset as the corresponding CPT samples from each track (Track 2 average time from symptom onset = 10 days; Track 3 average time from symptom

**Table 7. Statistical Correlation to Progression of Track 2 to Intubation.**

|  | Biomarker | Change | Estimate | Std. Error | P | OR | 95% CI of OR |
|---|---|---|---|---|---|---|---|
| Intubation | CRP | Day 3—Day 0 | 0.0062 | 0.0028 | 0.0277 | 1.0062 | (1.0007, 1.0118) |
|  | MIP-1β | Day 10—Day 3 | 0.1085 | 0.0552 | 0.0492 | 1.1146 | (1.0004, 1.2419) |
|  | MIP-1α | Day 10—Day 3 | 0.1092 | 0.0525 | 0.0366 | 1.1154 | (1.0062, 1.2363) |

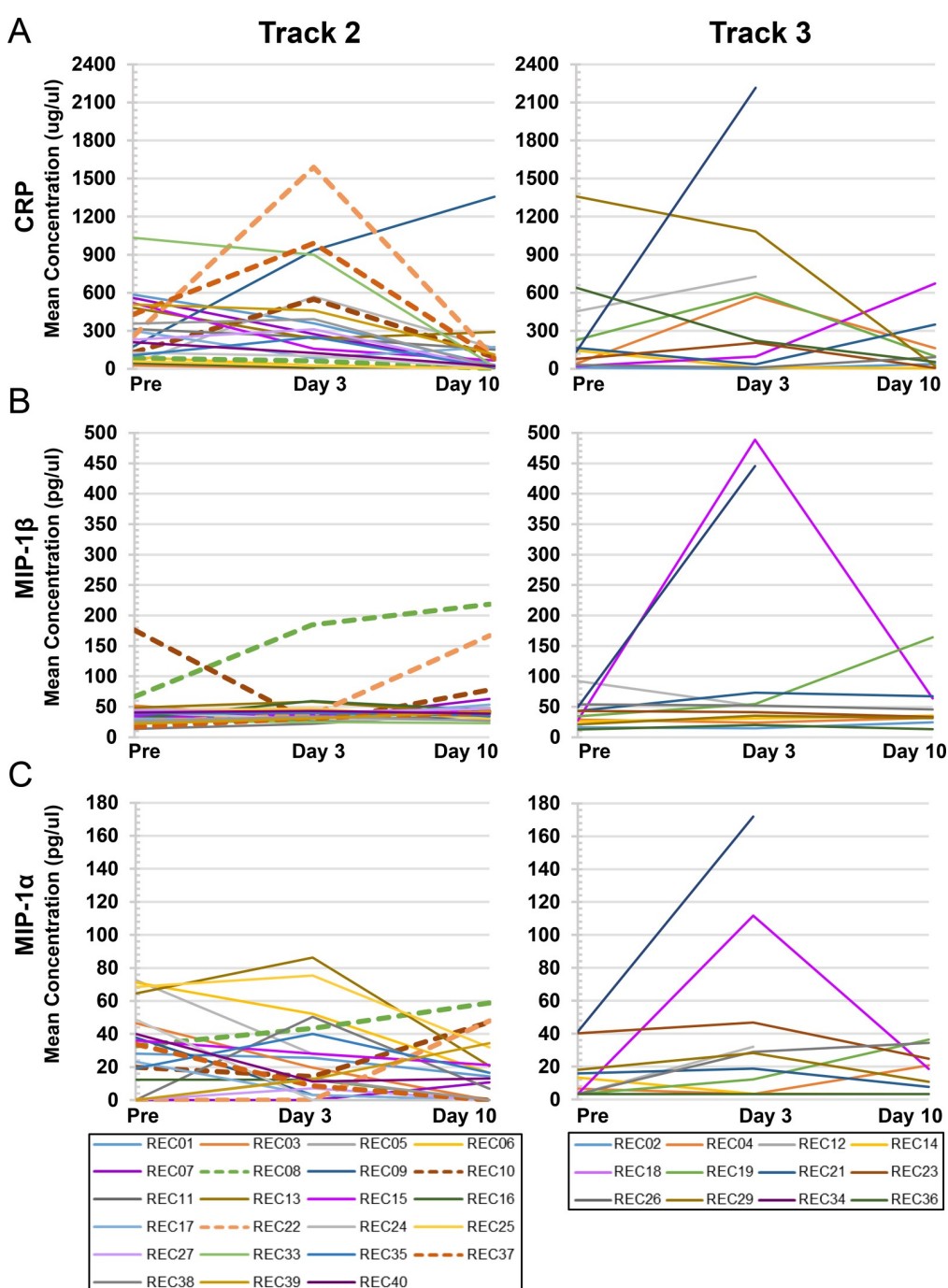

**Fig 5. Individual concentration plots of cytokines/chemokines correlating with progression to intubation in Track 2 recipients.** Statistical analyses, as detailed in Table 7, indicated that three analytes significantly correlated with the progression of COVID-19 respiratory disease in Track 2 recipients to the point of necessitating mechanical intubation (REC08, REC10, REC22, and REC37, indicated by dashed lines). Individual concentration plots are shown for recipients in Track 2 (left-hand panels; Pre-infusion, Day 3, n = 23; or Day 10, n = 22) and Track 3 (right-hand panels; Pre-infusion, Day 3, n = 12; or Day 10, n = 10), respectively for: (A) CRP; (B) MIP-1β; and (C) MIP-1α. The change in the mean concentration of CRP from Pre-Infusion to Day 3 correlated with progression to intubation (p = 0.028); for MIP-1β from Pre-Infusion to Day 3 (p = 0.049); and for MIP-1α from Day 3 to Day 10 (p = 0.037).

**Table 8. IgG and IgM SARS-CoV-2 Neutralizing Antibody Titer in Recipient Plasma.**

| Recipient | Pre-Infusion IgG Titer | Day 3 IgG Titer | Day 10 IgG Titer | Pre Infusion IgM | Day 3 IgM | Day 10 IgM |
|-----------|------------------------|-----------------|------------------|------------------|-----------|------------|
| REC01 | 500–1,000 | 1,000–10,000 | 100–500 | Neg | Pos | Pos |
| REC02 | 1,000–10,000 | >10,000 | >10,000 | ND | Pos | Pos |
| REC03 | >10,000 | ND | ND | Neg | ND | ND |
| REC04 | >10,000 | >10,000 | >10,000 | Pos | Pos | Pos |
| REC05 | >10,000 | >10,000 | >10,000 | Pos | Pos | Pos |
| REC06 | 1,000–10,000 | 1,000–10,000 | >10,000 | Pos | Pos | Pos |
| REC07 | 1,000–10,000 | 1,000–10,000 | 1,000–10,000 | Pos | Pos | Pos |
| REC08 | 1,000–10,000 | 1,000–10,000 | >10,000 | Pos | Pos | Pos |
| REC09 | **BLQ** | 100–500 | 1,000–10,000 | Neg | Pos | Pos |
| REC10 | **BLQ** | 100–500 | 100–500 | Neg | Neg | Neg |
| REC11 | **BLQ** | ND | 100–500 | Neg | ND | Neg |
| REC12 | 1,000–10,000 | >10,000 | ND | Pos | Pos | ND |
| REC13 | 100–500 | 1,000–10,000 | 1,000–10,000 | Neg | Pos | Pos |
| REC14 | 1,000–10,000 | >10,000 | 1,000–10,000 | Pos | Pos | Pos |
| REC15 | 100–500 | 1,000–10,000 | 1,000–10,000 | Pos | Pos | Pos |
| REC16 | 1,000–10,000 | 100–500 | >10,000 | Pos | Pos | Pos |
| REC17 | 100–500 | >10,000 | >10000 | Pos | Pos | Pos |
| REC18 | >10,000 | 1,000–10,000 | >10,000 | Pos | Pos | Neg |
| REC19 | 1,000–10,000 | >10,000 | >10,000 | Pos | Pos | Pos |
| REC21 | 500–1,000 | >10,000 | >10,000 | Pos | Pos | Pos |
| REC22 | 1,000–10,000 | 500–1,000 | >10,000 | Pos | Pos | Pos |
| REC23 | 1,000–10,000 | >10,000 | 1,000–10,000 | Pos | Pos | Pos |
| REC24 | 1,000–10000 | 1,000–10,000 | >10,000 | Neg | Pos | Pos |
| REC25 | 1,000–10000 | >10,000 | >10,000 | Neg | Pos | Pos |
| REC26 | >10000 | >10,000 | >10,000 | Pos | Pos | Pos |
| REC27 | 500–1000 | 1,000–10,000 | >10,000 | Pos | Pos | Pos |
| REC29 | 1,000–10,000 | 1,000–10,000 | >10,000 | Pos | Pos | Pos |
| REC33 | 100–500 | ND | >10,000 | Pos | ND | Pos |
| REC34 | 1,000–10,000 | >10,000 | ND | Pos | Pos | ND |
| REC35 | 1,000–10,000 | >10,000 | >10,000 | Pos | Pos | Pos |
| REC36 | >10,000 | >10,000 | >10,000 | Pos | Pos | Pos |
| REC37 | 100–500 | 1,000–10,000 | >10,000 | Pos | Pos | Pos |
| REC38 | BLQ | 100–500 | >10,000 | Neg | Neg | Pos |
| REC39 | 100–500 | 1,000–10,000 | >10,000 | Neg | Pos | Pos |
| REC40 | BLQ | 500–1,000 | BLQ | Neg | Neg | Neg |

onset = 15 days). The mean change in cytokine concentration (Δ) from Pre-infusion to Day 10 post-infusion in CPT samples or from the corresponding time-points in nonCPT patient samples (Track 2, ΔDay 10 to Day 20; Track 3 ΔDay 15 to Day 25) was calculated. The results are shown in **Fig 7**. In Track 2 patients, the change in concentration over time for several cytokines was significantly different between CPT and nonCPT patients. Several pro-inflammatory cytokines were decreased in patients who received CPT over the course of the 10-day study period, compared to nonCPT patients. These include IFNα2 (MeanΔ CPT, -31.53 ug/ml; MeanΔ nonCPT, 3.14 ug/ml p = 0.006), IFNγ (MeanΔ CPT, -48.36 ug/ml; MeanΔ nonCPT, -6.55 ug/ml p = 0.003), IL-12p40 (MeanΔ CPT, -37.51 ug/ml; MeanΔ nonCPT, 0.68 ug/ml p = 0.01), IL-17A (MeanΔ CPT, -7.769ug/ml; MeanΔ nonCPT, 2.07 ug/ml p = 0.04) and

**Table 9. Summary of Recipient Plasma Ig Isotype Concentrations Over Time.**

| Ig Isotype ± | Day | Range | Median | IQR | Mean | SD | % Elevated* | Ig Isotype ± | Day | Range | Median | IQR | Mean | SD | % Elevated* | P Value^ |
|---|---|---|---|---|---|---|---|---|---|---|---|---|---|---|---|---|
| | | | | **Track 2** | | | | | | | | **Track 3** | | | | |
| IgM | Pre-Inf | 29.46–1526.64 | 500.87 | 405.72 | 635.81 | 355.04 | 4.3 | IgM | Pre-Inf | 191.47–1810.14 | 545.92 | 504.78 | 651.91 | 464.52 | 8.3 | 0.460 |
| | 3 | 96.45–1377.92 | 614.29 | 387.09 | 607.07 | 328.10 | 0.0 | | 3 | 349.14–2085.05 | 552.00 | 368.72 | 691.10 | 488.77 | 8.3 | 0.300 |
| | 10 | 63.78–2043.42 | 868.66 | 565.72 | 921.22 | 497.52 | 13.6 | | 10 | 179.98–2821.53 | 464.56 | 312.31 | 711.68 | 767.45 | 10.0 | 0.220 |
| IgG1 | Pre-Inf | 467.84–3394.79 | 1508.58 | 1145.05 | 1686.83 | 763.37 | 0.0 | IgG1 | Pre-Inf | 598.72–5565.71 | 1574.16 | 1065.04 | 1958.64 | 1405.92 | 8.3 | 0.270 |
| | 3 | 633.39–4377.47 | 2538.12 | 1137.61 | 2467.19 | 892.29 | 0.0 | | 3 | 1215.2–5223.11 | 2320.29 | 1176.69 | 2692.29 | 1212.88 | 0.0 | 0.290 |
| | 10 | 471.80–6585.11 | 3930.08 | 1608.61 | 3805.31 | 1515.92 | 22.7 | | 10 | 1572.9–8232.79 | 2293.49 | 1498.74 | 3130.42 | 1963.08 | 10.0 | 0.180 |
| IgG2 | Pre-Inf | 557.28–7485.44 | 1592.47 | 1011.59 | 1836.37 | 1387.54 | 4.3 | IgG2 | Pre-Inf | <41.15–3593.54 | 1682.12 | 768.71 | 1783.23 | 802.90 | 16.7 | 0.440 |
| | 3 | 245.53–2398.19 | 1113.82 | 636.61 | 1151.99 | 505.30 | 0.0 | | 3 | 100.52–3137.44 | 1171.52 | 676.67 | 1193.21 | 800.61 | 8.3 | 0.440 |
| | 10 | 134.93–4841.17 | 1787.83 | 1014.11 | 1798.30 | 1126.12 | 13.6 | | 10 | 252.67–3685.76 | 1338.09 | 1366.52 | 1519.90 | 1100.62 | 20.0 | 0.260 |
| IgG3 | Pre-Inf | 6.53–2221.34 | 299.54 | 290.13 | 500.98 | 589.09 | 13.0 | IgG3 | Pre-Inf | 14.99–2119.72 | 280.21 | 259.16 | 396.59 | 563.87 | 8.3 | 0.310 |
| | 3 | 28.51–2952.21 | 530.21 | 533.48 | 656.51 | 716.05 | 13.0 | | 3 | 64.43–2204.29 | 425.58 | 479.70 | 539.44 | 582.10 | 8.3 | 0.300 |
| | 10 | 14.40–4081.00 | 811.41 | 2042.62 | 1409.24 | 1232.84 | 45.5 | | 10 | 76.30–3378.10 | 387.27 | 524.82 | 705.34 | 998.16 | 20.0 | 0.051 |
| IgG4 | Pre-Inf | 1.28–5284.39 | 300.04 | 598.12 | 1098.00 | 1771.44 | 26.1 | IgG4 | Pre-Inf | 0.51–1164.21 | 160.33 | 372.93 | 325.26 | 392.26 | 16.7 | 0.028 |
| | 3 | 15.36–6012.85 | 162.98 | 532.31 | 963.40 | 1833.56 | 17.4 | | 3 | 15.56–6955.33 | 224.07 | 1326.54 | 1457.64 | 2421.05 | 25.0 | 0.270 |
| | 10 | 5.34–9193.77 | 233.81 | 3142.14 | 1883.73 | 2942.47 | 36.4 | | 10 | 8.04–2810.92 | 177.82 | 606.66 | 657.25 | 969.69 | 30.0 | 0.045 |
| IgA | Pre-Inf | 55.41–1885.00 | 1102.55 | 843.81 | 976.95 | 485.37 | 4.3 | IgA | Pre-Inf | 36.17–3076.73 | 825.90 | 1251.37 | 1193.87 | 910.19 | 25.0 | 0.230 |
| | 3 | 127.47–1933.61 | 911.67 | 454.89 | 947.59 | 419.35 | 4.3 | | 3 | 156.41–2479.88 | 1153.34 | 798.39 | 1177.55 | 676.74 | 16.7 | 0.150 |
| | 10 | 85.53–3362.65 | 1279.64 | 670.32 | 1328.39 | 735.01 | 13.6 | | 10 | 568.51–2323.40 | 970.27 | 524.68 | 1163.83 | 612.29 | 10.0 | 0.260 |

±Unit is ug/ml

* The % of recipients with plasma values elevated above the normal control mean + 2xSD

^T-test was used to compare Means of Track 2 and Track 3; Statistically significant p values are highlighted in gray

SD, Standard Deviation of the Mean; IQR, Interquartile Range; Pre-Inf, Pre-Convalescent Plasma Infusion

RANTES (MeanΔ CPT, -2777.75 ug/ml; MeanΔ nonCPT, 684.95 ug/ml p = 0.025). In both Track 2 and Track 3 patients, NGAL levels increased more significantly in CPT patients compared to nonCPT patients (Track 2, 1346.02 ug.ml vs 425.73 ug/ml, p = 0.004; Track 3, 1557.32 ug/ml vs 46.86 ug/ml, p = 0.03). Also notable was the difference in the change in IL-1RA levels between CPT and nonCPT patients in Track 3 (MeanΔ CPT, 78.83 ug/ml; MeanΔ nonCPT, -44.77 ug/ml p = 0.03). The remaining data which did not reach statistical analysis is shown in **S1 Fig**.

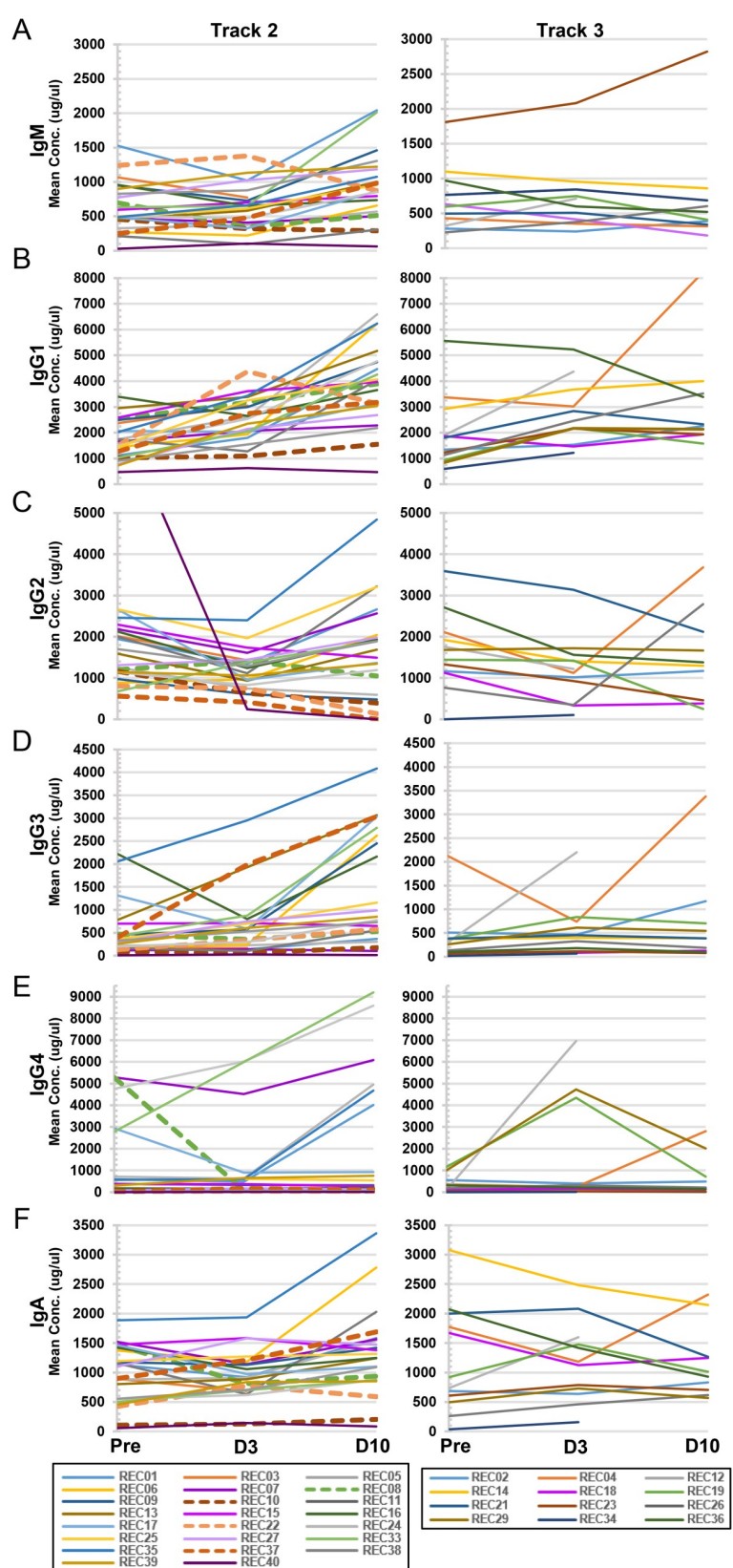

**Fig 6. Individual concentration plots of plasma Ig isotypes in Track 2 and Track 3 recipients.** Plasma concentration levels of Ig isotypes from individual recipients were measured and plotted relative to Track 2 (left-hand panels; Pre-infusion, Day 3, n = 23; or Day 10, n = 22) or Track 3 (right-hand panels; Pre-infusion, Day 3, n = 12; or Day 10, n = 10) cohorts. Displayed are: (A) IgM; (B) IgG1; (C) IgG2; (D) IgG3; (E) IgG4; and (F) IgA. Dashed lines indicate Track 2 patients who progressed to intubation post-infusion.

## Discussion

As the global SARS-CoV-2 pandemic continues to rage, there is controversy over whether convalescent plasma transfers provide effective benefit to patients hospitalized with COVID-19 and associated pneumonia. Our institution at Hackensack Meridian Health conducted a phase IIa clinical trial to test the safety and efficacy of CPT from donors with high titers of SARS-CoV-2 neutralizing antibodies. This trial, which is independent of the national Mayo Clinic

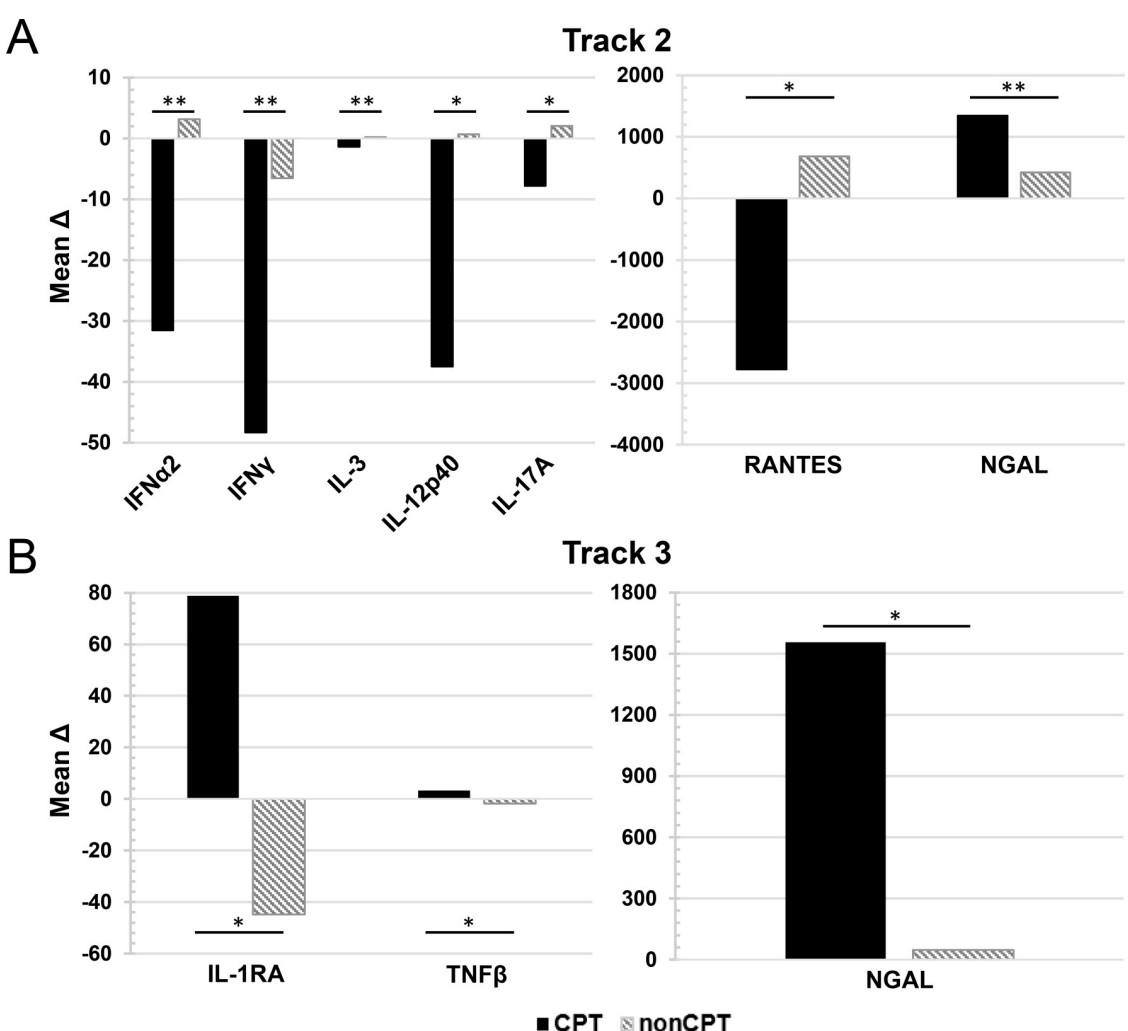

**Fig 7. Mean change in concentration from pre-infusion to Day 10 post-infusion of patients with and without CPT.** The mean change (Δ) in cytokine concentration between pre-infusion levels (Day 0) and Day 10 post-infusion for CPT patients and corresponding10-day period for nonCPT treated patients were calculated in patients categorized as Track 2 (A) and Track 3 (B). Statistically significant changes in mean values between those patient who received CPT (Track 2, n = 22; Track 3, n = 10) and those who did not (nonCPT; Track 2, n = 8; Track 3, n = 10) were evaluated via T-test. Significance is denoted by (*) for $0.01 \leq p \leq 0.05$ or (**) for $p < 0.01$.

Expanded Access Protocol [17], demonstrated improved overall survival (89.5%) and less requirement for intubation of patients who initially entered the hospital without mechanical ventilation [21]. To better understand the scientific basis for observed improvements, we analyzed the blood plasma concentration of 35 cytokines and chemokines, as well as Ig isotypes, to investigate elevations above normal controls and to note changes in the first 10 days after infusion of the donor plasma.

First, our cytokine/chemokine analysis of donor plasma, itself, revealed that elevated levels of IFN-α2, IL-6, PCT, and CRP were present, ranging from 20.0–31.4% of donors. Upregulation of IL-6 has been reported in COVID-19 patients and is a contributing factor to cytokine storm and the development of ARDS [8,23–27]. PCT, often associated with bacterial infections and tissue injury has also been correlated with severe COVID-19 patients [27,28]. CRP, which is commonly found as a result of inflammation and tissue damage has also been observed to be upregulated in COVID-19 patients [26,27]. The question is raised whether the transfer of plasma containing elevated levels of these cytokines could aggravate or even accelerate the development of ARDS or a more severe form of COVID-19 disease in their recipients? Consistent with that possibility, two of the donor plasmas had elevations in at least three of the four cytokines mentioned above, Don10/Don11 (**Table 2**). The recipient of Don10 (REC10) was in Track 2 but developed ARDS and organ dysfunction, progressed to intubation, and ultimately succumbed (**Table 3**). Don29 had elevations in all four of the cytokines. REC29, who was in Track 3, also developed ARDS, but was able to survive. A more focused study on this question would need to be performed to determine if exclusion of donor plasma with multiple cytokine elevations might be appropriate as an extra precaution.

The other interesting observation of donor plasma was the apparent class switch in Ig isotypes towards IgG4, with high elevations occurring in 34% of the donor plasma units (**Table 2**). Although it is not uncommon for Ig class switching to occur over time, this is the first report of a significant presence of IgG4 in convalescent COVID-19 plasma donors. Elevated IgG4 is at the root of IgG4-related systemic diseases, including cardiovascular diseases, and is pathogenic in some autoimmune disease [29,30]. However, IgG4 can also dampen immune responses by competing with more effective Ig class antibodies and the fact that it does not bind complement C1q nor bind well to Fc-gamma- receptors on myeloid cells, which are important for opsonization of infectious agents [31]. In this regard, it is worth noting that in respiratory viral infections, type 2 conventional dendritic cells (cDC1s and cDC2s), in addition to monocytes, in the lungs acquire expression of Fc receptors and are key to presenting antigen to responding CD4 T cells [32]. Therefore, the presence of IgG4, although it would still bind to virus particles, could potentially diminish immune activation in response to SARS-CoV-2. The impact of elevated IgG4 in SARS-CoV-2 infection will require further exploration.

In regard to the anti-SARS-CoV-2 neutralizing IgG titers, we initially postulated that selecting convalescent plasma donors with high titers would optimize the benefit to the recipients, although our study was not designed to evaluate optimal donor neutralizing antibody titers. Classical pharmacology recognizes the importance of delivering a drug at or above its pharmacodynamic target level. The delivery of high titer neutralizing antibodies for CPT is now recognized as crucial to successful clinical outcomes [20]. This approach may serve to help standardize the quality of the donor product, particularly in light of growing evidence that anti-viral neutralizing antibody and immune responses may wane with protracted time after COVID-19 infection [33–35].

Longitudinal blood plasma sampling of patients during the course of CPT treatment allowed for the analysis of cytokine/chemokine changes over time in this patient cohort. Several analytes were found to be elevated in patients prior to infusion with convalescent plasma

(Fig 2). Consistent with several other reports, we found a significant proportion of patients' blood plasma samples (>50%) with elevated levels of IL-6, IL-7, IL-8, IP-10, CRP, MCP-1, and MIP-1β [2,8,23,32,36–38]. Elevations in these cytokines are consistent with the immune dysfunction reported in COVID-19 patients, including B cell and myelomonocytic composition changes and altered T cell phenotypes [38]. The percentage of patients' plasma samples with elevations in IL-1RA and NGAL increased significantly over the course of 10 days for patients categorized in both Track 2 and Track 3, while the percentage of patients with elevated blood plasma EGF and VEGF rose steadily only in Track 2. This is also reflected in the mean concentrations with the concentration of both EGF and VEGF rising significantly in Track 2 patients' samples over the 10-day course, while concentrations of these cytokines remained steady in the Track 3 cohort. These results align with those reported by Lucas et al, in which cluster analysis identified 4 distinct immune profiles, one of which included high levels of growth factors such as EGF, VEGF, and IL-7 [39]. They found that patients with moderate disease were enriched for cytokines with this growth factor signature. This finding aligns with our findings of significant elevations of these growth factors in the blood plasma of Track 2 patients compared to Track 3.

We also observed a significant decrease in the percentage of patients' blood plasma samples with elevated cytokines by day 10 in Track 2 patients including IFN-α, IFN-γ, IL-3, IL-7, IL-12p40, IL-12p70, IL-13, IL-17A, TNF-β, and CRP. This aligned with the significant decreases in mean concentrations of these cytokines observed over time, barring IL-7 and TNF-β. It is important to note that these cytokines are involved in both type 1 (Th1), type 2 (Th2), and type 3 (Th17) responses. On the contrary, for patient samples in Track 3, only RANTES was noted to have a significant decrease in the percentage of patients with elevations over time. This was also reflected in the decrease in mean concentration of this cytokine in the Track 3 cohort. Other longitudinal analyses of cytokines in COVID patients have shown similar results [23,39,40]. Lucas et al. reported a steady decline in a similar immune activation signature in patients with moderate disease with minimal changes in patients with severe disease [39]. Zhao et al. have previously reported an association of RANTES with mild disease [40]. However, in our study, we observed a greater percentage of patients with more severe disease (66.7%, Track 3) with elevated RANTES compared to those in Track 2 (26.1%) at the Pre-Infusion time point. It is important to note that these previously reported studies measured changes in cytokines beginning at symptom onset, whereas our measurements began prior to the start of CPT infusion which varied in our patient cohort from 2–27 days post-symptom onset. Differences in results may also be due to the clinical definition of mild, moderate and severe disease.

The design of our study enabled us to search for associations between changes in cytokine/chemokine blood plasma concentrations over time and defined clinical endpoints for patients in Track 2 and Track 3. This analysis identified increases in three analytes that were associated with progression to intubation in Track 2 patients—CRP, MIP-1α, and MIP-1β, albeit effect sizes were small, but significant (Fig 5). While the majority of patients in Track 2 saw levels of CRP declining over time or rising only minimally, 3 of the 4 patients who progressed to intubation had a sharp increase in CRP at Day 10. CRP has been well reported to be a prognostic indicator of disease severity and respiratory decline in COVID-19 patients [41–43]. In this analysis, no cytokine/chemokine changes were found statistically to be associated with mortality by Day 30 post-CPT, the clinical endpoint in our Track 3 cohort.

In order to examine if patients receiving CPT exhibited different cytokine kinetics compared to those patients who did not receive CPT, we obtained plasma samples from our institutional biorepository of hospitalized COVID-19 patients in Track 2 and Track 3 who received standard care. In order to compare changes in cytokine levels over the 10-day study period,

the mean change (Δ) in cytokine concentration was calculated. Several cytokines were found to display significantly different kinetics in CPT patients compared to nonCPT patients during the 10-day period. Notably, several inflammatory cytokines were found to have a larger mean decrease over the 10-day period in patients who received CPT compared to those who did not, particularly in Track 2 patients. These included IFNα2, IFNγ, IL-12p40, IL-17A, and RANTES. Our analysis was limited by a small sample size. In addition, the nonCPT patient samples were analyzed retrospectively and patient characteristics were not available. While these samples were collected from within our hospital system during the same time-frame as the CPT patient samples, a randomized, prospective trial would need to be performed to confirm these preliminary findings. It is known that the inflammatory cytokine milieu triggered by SARS-CoV-2 infection contributes to disease progression, ARDS, and death in hospitalized patients. While more research is needed, our results indicate that CPT may contribute to the resolution of this inflammatory response.

This study was limited by a small sample size resulting in a lack of statistical power to identify minor differences in mean cytokine and Ig isotype concentrations. In addition, the initial design of our phase IIa trial lacked a randomized control group of patients hospitalized with COVID-19 pneumonia that did not receive CPT. While we were able to retroactively obtain and analyze cryopreserved plasma samples from nonCPT patients, the conclusions that can be drawn from these analyses are limited. The time frame of disease onset to CPT infusion ranged greatly in our patient cohort from 2–27 days. This could account for some of the variability observed in cytokine/chemokine and Ig plasma concentrations. Due to this large variability, in addition to presenting mean concentrations, we also present our results as the percentage of patients with elevations in cytokine/chemokine/Ig concentrations and have defined those analytes exhibiting elevations in greater than 20% of patients as noteworthy. Presenting the data in this fashion impedes direct comparisons with some published reports. It is also important to note that while the patient characteristics between Track 2 and Track 3 patients were comparable in most categories, there was a significant difference between Track 2 and Track 3 patients in regards to the administration of corticosteroids and tociluzumab, with these treatments used more frequently in the Track 3 cohort (**Table 3**). This is not surprising given the more severe clinical condition of Track 3 patients, however, it is possible that these treatments could impact the production of cytokines/chemokines and needs to be considered when evaluating the observed differences between Track 2 and Track 3 patients in our study.

In conclusion, this study of inflammatory cytokine/chemokine and Ig isotype elevations in the blood plasma of hospitalized COVID-19 patients that were administered convalescent plasma therapy has revealed various patterns of increases and declines within the 10-day study period, and some notable differences between patients that were or were not mechanically ventilated at time of infusion. These correlative measures were part of the initial Phase IIa clinical trial which did support a benefit of the therapy with hospitalized patients early in the disease course. We intend to follow up with analyte measures in the next Phase II randomized trial which is currently enrolling for high titer convalescent plasma therapy in COVID-19 test positive individuals within four days of symptom onset. This will include a control arm of similarly positive individuals who will not receive CPT and therefore will help us to determine whether the treatment actually tempers the elevations and duration of inflammatory cytokine/chemokines.

## Supporting information

**S1 Fig. Mean change in concentration from pre-infusion to Day 10 post-infusion of patients with and without CPT.** The mean change (Δ) in cytokine concentration between

pre-infusion levels (Day 0) and Day 10 post-infusion for CPT patients and corresponding10-day period for nonCPT treated patients were calculated in patients categorized as Track 2 (A) and Track 3 (B). Statistically significant changes in mean values between those patient who received CPT (Track 2, n = 22; Track 3, n = 10) and those who did not (nonCPT; Track 2, n = 8; Track 3, n = 10) were evaluated via T-test. Significance is denoted by (*) for $0.01 < p < 0.05$ or (**) for $p < 0.01$.
(TIF)

**S1 Table. Donor Plasma Luminex Analyses; Concentration (pg/ml) of Analytes with Elevation 0–19% of Donors.**
(DOCX)

**S2 Table. Recipient Plasma Luminex Analyses; Pre-Infusion Concentration (pg/ml) of Analytes with Elevation 0–19% of Recipients.**
(DOCX)

**S3 Table. Recipient Plasma Luminex Analyses; Day 3 Concentration (pg/ml) of Analytes with Elevation 0–19% of Recipients.**
(DOCX)

**S4 Table. Recipient Plasma Luminex Analyses, Day 10 Concentration (pg/ml) of Analytes with Elevation 0–19% of Recipients.**
(DOCX)

**S5 Table. Plasma Concentration of Analytes in Normal Controls.**
(DOCX)

**S6 Table. Statistical Analyses of Comparison Between Time Points of Mean Concentrations of Cytokines/Chemokines.**
(DOCX)

**S7 Table. Pre-Infusion Individual Recipient Cytokine Levels of Interest.**
(DOCX)

**S8 Table. Day 3 Individual Recipient Cytokine Levels of Interest.**
(DOCX)

**S9 Table. Day 10 Individual Recipient Cytokine Levels of Interest.**
(DOCX)

**S10 Table. Individual Recipient Plasma Ig Isotype Concentrations Over Time.**
(DOCX)

## Author Contributions

**Conceptualization:** Robert Korngold, David S. Perlin.

**Formal analysis:** Ming Tan, Bingsong Zhang.

**Funding acquisition:** Michele L. Donato, David S. Perlin.

**Investigation:** Stacey L. Fanning, Zheng Yang, Kira Goldgirsh, Steven Park.

**Methodology:** Stacey L. Fanning.

**Project administration:** Robert Korngold.

**Resources:** Steven Park, Joshua Zenreich, Melissa Baker, Phyllis McKiernan, Michele L. Donato.

**Supervision:** Robert Korngold.

**Visualization:** Stacey L. Fanning, Robert Korngold.

**Writing – original draft:** Stacey L. Fanning, Robert Korngold.

**Writing – review & editing:** David S. Perlin.

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
