## [Decision Letter · Decision Letter 0]

24 Jul 2021

Dear Dr. Fanning,

Thank you very much for submitting your manuscript "Elevated Cytokines and Chemokines in Peripheral Blood of Patients with SARS-CoV-2 Pneumonia Treated with High-Titer Convalescent Plasma" for consideration at PLOS Pathogens. As with all papers reviewed by the journal, your manuscript was reviewed by members of the editorial board and by several independent reviewers. In light of the reviews (below this email), we would like to invite the resubmission of a significantly-revised version that takes into account the reviewers' comments.

We cannot make any decision about publication until we have seen the revised manuscript and your response to the reviewers' comments. Your revised manuscript is also likely to be sent to reviewers for further evaluation.

Sincerely,

Shin-Ru Shih

Section Editor

PLOS Pathogens

Shin-Ru Shih

Section Editor

PLOS Pathogens

Kasturi Haldar

Editor-in-Chief

PLOS Pathogens

orcid.org/0000-0001-5065-158X

Michael Malim

Editor-in-Chief

PLOS Pathogens

orcid.org/0000-0002-7699-2064

Reviewer's Responses to Questions

**Part I - Summary**

Reviewer #1: This was a single institution prospective phase II clinical trial which involved 51 patients who received treatment by using donor plasma of high neutralizing titer anti-SARS-CoV-2 (data have been published), and investigated the levels of peripheral blood cytokines, chemokines, and Ig isotypes from the first 35 patients enrolled in the original study in the first 10 days after infusion. Plasma recipients were divided into hospitalized COVID-19 pneumonia patients but who did not (Track 2) or did (Track 3) require mechanical ventilation. They found that several cytokines were elevated in the patients of each Track and some continued to rise through Day 10, while others initially increased and then subsided. Furthermore, elevations in MIP-1α, MIP-1β and CRP correlated with disease progression of Track 2 recipients.

Reviewer #2: This is an original study examining the cytokine, chemokine and immunoglobulin (Ig) response in a cohort of individuals who had received convalescent plasma transfer (CPT) during the 1st wave of the pandemic in the spring of 2020 at a single medical center as part of a phase 2a prospective clinical trial. Overall, this is a well written manuscript with novel observations from extensive profiling of cytokine/chemokine, Ig of donor plasma and recipients. Findings of elevated cytokine/chemokine in convalescent donors, and Ig class switching to IgG4 are interesting and hypothesis generating especially as we are observing long term persistent symptoms in some of the convalescent individuals. Additionally, the elevated cytokine/chemokine levels in the recipients suggest that the enrolled cohort may have been in the hyper-inflammatory phase as opposed to the viral replicatory phase where CPT is traditionally thought to be effective in. It would be important to understand if CPT have additional immunomodulatory role in those with advanced disease. Main weakness of the study is it is not clear how the main findings of cytokine/chemokine changes by tracks relate with CPT and what the author’s hypothesis is regarding the association. It is not clear how the overall findings of cytokine/chemokine changes will help guide therapeutics.

**Part II – Major Issues: Key Experiments Required for Acceptance**

Reviewer #1: 1. It will be much more clearly and convinced if the tables characterizing the patients were divided into two groups: track2 and track 3, and calculated the P value.

2. For the two groups (Track 2 and Track 3), the infusion titers of neutralizing anti-SARS-CoV-2 antibodies were the same or not?

3. Other treatments were same or not for the two groups? Especially anti-viral drug, corticoids? How about the rates of comorbidity for the two groups?

4. “for either a planned fresh infusion of 500 mL or for cryopreservation in aliquots of 200 mL”, this might make differences.

5. For better comparison, if the detection data of track2 and 3 were put in the same figures.

6. The number of patients were limited to draw the conclusion.

Reviewer #2: 1. Limitation section should further discuss the lack of a control group with corresponding cytokine/chemokine and Ig levels that limits establishing any causality between CPT and change in the measured levels. The findings such as elevated IL-6 and CRP may be an expected findings of severe Covid-19 and no inference can be made regarding the effect of CPT or how the findings can be utilized as biomarkers in using CPT. The effect size of the logistic regression finding was small despite statistical significance limiting clinical meaningfulness of the finding. Additionally, some of the relevant clinical variables that may be confounders were not taken into account (such as age, sex, BMI, concomitant medications) and small sample size may have limited further analysis. To gain more insight into the association between CPT and the change in cytokine/chemokine levels in this single arm study, it may be interesting to analyze by CP titers or baseline serostatus of CP recipients if sample size allows.

2. The presentation of the data may be improved by revising the formats of the tables. Alternative to presenting raw individual data, it may be helpful to present the descriptive findings as median (IQR) or mean (SD) depending on the distribution for continuous variables and freq (%) for categorical variables.

3. In the 2nd and 3rd paragraph of the discussion section, authors make speculative statements regarding possible transfer of inflammatory cytokine/chemokines from CP unit to the recipients and the effect of IgG4 in CP unit that are not sufficiently supported by the data presented. Causality is inferred from their observations of cytokine/chemokine levels in small number of CP units and recipients’ plasma and suggest possible transfer of the cytokines from CP unit to recipient. Additionally, this hypothesis was not clearly prefaced in introduction or methods. It may help to clarify in the introduction section if the authors are suggesting that CPT had an immunomodulatory effect or caused the elevation of cytokine/chemokines associated with cytokine storm. Statement in line 415, 416 “exclusion of donor plasma with multiple cytokine elevations might be appropriate as an extra precaution” seem to be relatively strong without sufficient supporting data.

**Part III – Minor Issues: Editorial and Data Presentation Modifications**

Reviewer #1: Page 7, line 44, “A phase IIb clinical trial”, Page 11, line 116, “phase IIa clinical trial”, IIa or IIb?

Reviewer #2: 1. Some of the results are presented in the introduction and discussion section. For example, Page 6, lines 105-108 may fit better in the results section.

2. Consider reporting the ‘n’ in the figures when available, presence of any missing samples, and if blood samples were drawn in all participants or only those who remained hospitalized at day 10.

3. Page 5, Line 91, 92: Consider providing details on how the non-CPT treated subjects in the original study were identified – whether they were from a randomized prospective study or were retrospectively identified control group.

4. Page 5, Line 92: Can state P=0.036 instead of P<0.036?

5. Page 6, line 110-112: It is not clear what the authors mean that identifying biomarkers will improve outcome in the pre- and post-vaccination period.

6. page 10, line 184. It is not clear why there is a range, 12-16, of healthy adult donors. Additionally, would clarify if the healthy donors were screened for SARS-CoV-2 and were negative.

7. page 10, line 185-186. What is the criterion used for elevation of correlates (ie. values greater than 2 SD from the mean of the healthy controls) based on? Is it a standard definition?

8. page 10, line 191. It is not clear how progression to positive pressure mechanical ventilation was measured. If it is delineated in the original paper, would include the reference.

9. page 10, line 203: Would clarify if mean (SD) and t test were used based on normal distribution of the data. Consider including statistical program used for completeness.

10. page 12, line 208-213: Details regarding the donors in the result section seem repetitive as that was presented in the methods section.

11. page 12, line 212. Is “infectious disease markers” a standardized test for pathogens that can be transmitted by transfusion?

12. page 13, line 233. It is not clear which font is marked in red. Is it referring to the values highlighted in grey in table 2A?

13. Recommend reformatting table 3, patient characteristics. Continuous variables can be presented as median (IQR) and categorical variables as freq (%). Consider presenting the number of days from admission to transfusion which can give an insight on how early in admission the patients were transfused.

14. Race variable is categorized as Caucasian, African American, Hispanic and Asian. Hispanic vs non Hispanic should be included in a ethnicity variable. Consider making Race/Ethnicity variable and categorize as non-hispanic white, non-hispanic black, Hispanic etc.

15. page 14, line 251. Can include measure of dispersion (IQR) in addition to median.

16. page 14, lines 253-254. Repetitive to have definitions of track 2 and 3 reported again.

17. page 22, line 433, 434. Authors state, “in regard to the anti-SARS-CoV-2 neutralizing IgG titers, we initially hypothesized that selecting CP donors with high titers would optimize the benefit to the recipients”. It is not clear if this dose response was examined and if this initial hypothesis was supported by the author’s data or if this is a general statement. May consider editing the sentence slightly to clarify.

PLOS authors have the option to publish the peer review history of their article (what does this mean?). If published, this will include your full peer review and any attached files.

Reviewer #1: No

Reviewer #2: **Yes: **Hyunah Yoon
---

## [Editor Report · Decision Letter 1]

11 Oct 2021

Dear Dr. Fanning,

We are pleased to inform you that your manuscript 'Elevated Cytokines and Chemokines in Peripheral Blood of Patients with SARS-CoV-2 Pneumonia Treated with High-Titer Convalescent Plasma' has been provisionally accepted for publication in PLOS Pathogens.

Best regards,

Shin-Ru Shih

Section Editor

PLOS Pathogens

Shin-Ru Shih

Section Editor

PLOS Pathogens

Kasturi Haldar

Editor-in-Chief

PLOS Pathogens

orcid.org/0000-0001-5065-158X

Michael Malim

Editor-in-Chief

PLOS Pathogens

orcid.org/0000-0002-7699-2064
---

## [Editor Report · Acceptance letter]

22 Oct 2021

Dear Dr. Fanning,

We are delighted to inform you that your manuscript, "Elevated Cytokines and Chemokines in Peripheral Blood of Patients with SARS-CoV-2 Pneumonia Treated with High-Titer Convalescent Plasma," has been formally accepted for publication in PLOS Pathogens.

Best regards,

Kasturi Haldar

Editor-in-Chief

PLOS Pathogens

orcid.org/0000-0001-5065-158X

Michael Malim

Editor-in-Chief

PLOS Pathogens

orcid.org/0000-0002-7699-2064